# MatLLMSearch: Crystal Structure Discovery with Evolution-Guided Large Language Models

## Abstract

Crystal structure generation is fundamental to materials science, enabling the discovery of novel materials with desired properties. While existing approaches leverage Large Language Models (LLMs) through extensive fine-tuning on materials databases, we show that pre-trained LLMs can inherently generate novel and stable crystal structures without additional fine-tuning. Our framework employs LLMs as intelligent proposal agents within an evolutionary pipeline that guides them to perform implicit crossover and mutation operations while maintaining chemical validity. We demonstrate that MatLLMSearch achieves a 78.38% metastable rate validated by machine learning interatomic potentials and 31.7% DFT-verified stability, outperforming specialized models such as CrystalTextLLM. Beyond crystal structure generation, we further demonstrate that our framework adapts to diverse materials design tasks, including crystal structure prediction and multi-objective optimization of properties such as bulk modulus, all without fine-tuning. These results establish our framework as a versatile and effective framework for consistent high-quality materials discovery, offering training-free generation of novel stable structures with reduced overhead and broader accessibility.

## 1 Introduction

Crystal Structure Generation (CSG) and Prediction (CSP) represent critical bottlenecks in materials discovery, requiring both chemical validity and thermodynamic stability to determine whether a material can be synthesized (Bagayoko, 2014). These tasks demand navigating an expansive chemical space while satisfying multiple constraints: three-dimensional periodicity, proper atomic coordination, charge balance, and minimized formation energy. While computational approaches have emerged as indispensable tools for accelerating materials discovery (Dunn et al., 2020; Eremin et al., 2023), developing reliable systems that effectively explore this vast and complex space remains challenging.

Recent advances in deep learning have introduced various approaches for structure prediction, from variational autoencoders to diffusion models (Flam-Shepherd and Aspuru-Guzik, 2023; Gruver et al., 2024; Jiao et al., 2024; Xie et al., 2022; Zeni et al., 2025). Meanwhile, Large Language Models (LLMs) have emerged as powerful tools for materials discovery (Achiam et al., 2023; Antunes et al., 2023; Fu et al., 2023). Prior work (Flam-Shepherd and Aspuru-Guzik, 2023) demonstrated that autoregressive models using character-level tokenization can generate valid crystal structures, and Gruver et al. (2024) showing that fine-tuning pre-trained language models like Llama (Grattafiori et al., 2024) can produce physically stable structures.

Current approaches often fine-tune LLMs on materials databases such as the Materials Project (Gruver et al., 2024), which contains only tens of thousands of structures compared to the vast space of possible stable compounds. While we propose a fundamentally different perspective: Recognizing pre-trained LLMs not as tools requiring domain-specific fine-tuning, but as intelligent agents already possessing rich embedded knowledge from vast scientific corpora. This perspective raises the question: *How can we exploit the comprehensive scientific knowledge already embedded in pre-trained LLMs to build a system that can consistently produce valid stable crystal structures?*

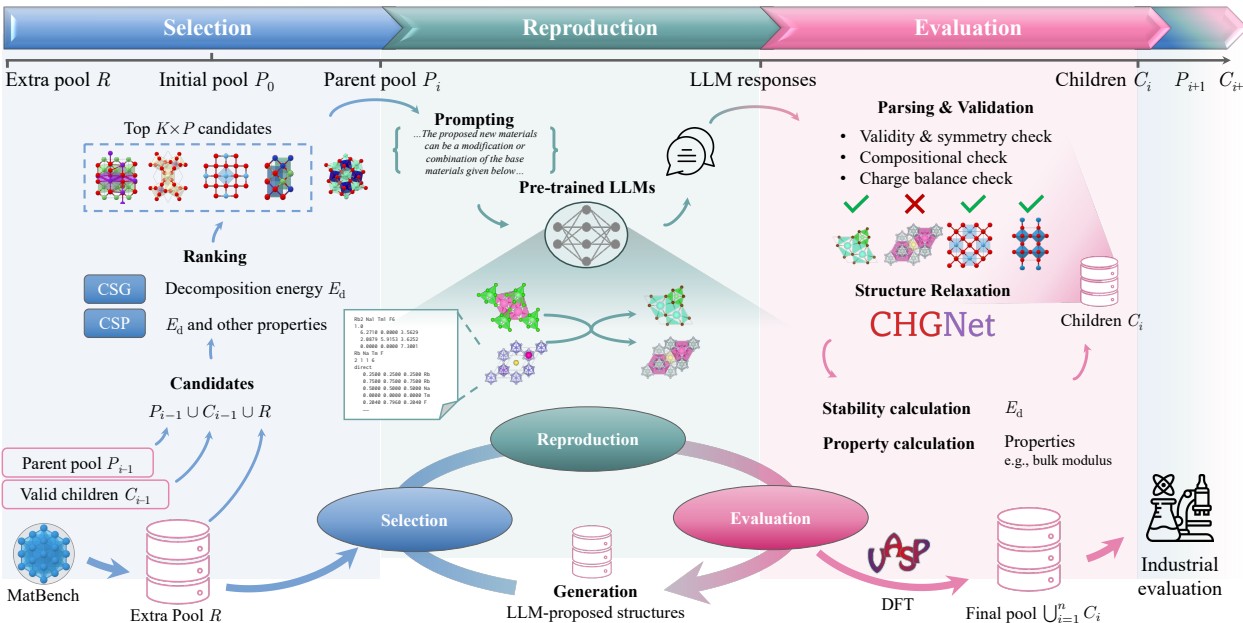

Figure 1: The workflow of MatLLMSearch for crystal structure generation. Starting from an initial population of known structures, our framework iteratively evolves new crystal structures through LLM-guided reproduction, evaluation, and selection.

Intuitively, we may directly prompt a commercial LLM to generate crystal structures. However, our ablation study in Section 4.4 across multiple configurations reveal that simple prompting fails to consistently generate valid crystal structures that are both stable and novel. These attempts often produce either copies of known structures, chemically invalid configurations, or thermodynamically unstable structures. The failure suggests that LLMs struggle to simultaneously satisfy the multiple constraints of CSG, indicating the need for a more sophisticated approach to exploit the potential of LLMs for materials discovery.

Evolutionary algorithms provide an effective framework for exploring the vast chemical space (Allahyari and Oganov, 2020; Oganov and Glass, 2006; Wang et al., 2025). By mimicking biological evolution through iterative selection, reproduction, and mutation operations, these algorithms can gradually improve the candidates, enabling automated property-guided materials optimization. Previous evolutionary approaches to CSG and CSP rely on explicit crossover and mutation operators, such as swapping structural motifs or introducing atomic displacements. While effective, these traditional operators lack the chemical intuition to efficiently navigate the complex constraints of crystal structures, often resulting in physically implausible candidates.

Our work advances this paradigm by leveraging the rich scientific knowledge embedded in LLMs to perform chemically-informed operations within the evolutionary algorithm framework. Unlike traditional operators that manipulate structures based on predefined rules, LLMs can implicitly reason about chemical bonding patterns, structural motifs, and stability principles learned from vast scientific literature (Boiko et al., 2023; Bran et al., 2024; Guo et al., 2023). This knowledge-guided approach enables more intelligent exploration of the chemical space, potentially discovering novel structures that traditional evolutionary methods might miss due to their limited chemical knowledge.

In this work, we introduce MatLLMSearch, a novel framework that integrates the rich scientific knowledge of LLMs into the evolutionary framework for crystal structure discovery. In our proposed framework, LLMs function as intelligent proposal agents that analyze parent structures to perform implicit crossover and mutation operations, while Machine Learning Interatomic Potentials (MLIPs) evaluate the physicochemical validity of generated structures. As illustrated in Figure 1, through iterative selection, reproduction, and evaluation, MatLLMSearch progressively discovers crystal structures with desired properties.

Our experiments demonstrate that MatLLMSearch successfully generates diverse, thermodynamically stable crystal structures while maintaining crystallographic validity. The framework achieves a 76.8% metastable structure generation rate, with 31.7% of structures verified as stable through DFT calculations, surpassing the state-of-the-art fine-tuned model CrystalTextLLM (Gruver et al., 2024). Notably, this performance is achieved with minimal computational overhead, requiring only LLM inference and stability evaluation with MLIPs rather than extensive model training.

Beyond crystal structure generation, our framework demonstrates remarkable flexibility across various materials discovery tasks. Through simple modifications in prompting and reference seed structures selection, our method extends to CSP, which we validate by discovering several metastable $Na_3AlCl_6$ polymorphs with significantly higher stability than existing structures in the Materials Project database. Furthermore, the framework enables multi-objective optimization of properties such as bulk modulus, without requiring specialized fine-tuning. While we demonstrate results using general-purpose pre-trained LLMs, the framework could also incorporate domain-specialized fine-tuned models or alternative search algorithms, offering a computationally efficient approach to materials discovery with reduced overhead and broader accessibility.

## 2 Background: Computational Materials Discovery with Machine Learning

### 2.1 Problem Definition

**Crystal Structure Generation (CSG).** The objective of CSG is to learn a probability distribution $p(c, l, s)$ over crystalline materials, where $c \in \mathbb{R}^{N \times M}$ represents the chemical composition matrix for $N$ atoms of $M$ distinct chemical species, $l \in \mathbb{R}^6$ denotes the lattice parameters (lengths and angles), and $s \in \mathbb{R}^{N \times 3}$ defines the spatial coordinates of atoms within a periodic unit cell. Samples drawn from this distribution should ideally satisfy fundamental thermodynamic stability criteria.

**Crystal Structure Prediction (CSP).** CSP addresses a more constrained problem of determining stable crystal structures for a specified chemical composition. Formally, it learns a conditional probability distribution $p(s, l \mid c)$ to identify thermodynamically favorable atomic arrangements and lattice parameters given a fixed composition $c$. This formulation addresses the practical scenario of discovering stable polymorphs for a specified chemical formula.

**Crystal Structure Design (CSD).** CSD extends beyond structure prediction by incorporating property optimization and conditional generation. An example objective is finding the optimal crystal structure that maximizes a target property $h(c, l, s)$: $m^* = \text{argmax}_{c,l,s \sim p(c,l,s)} h(c, l, s)$, where $h : \mathbb{R}^{N \times K} \times \mathbb{R}^6 \times \mathbb{R}^{N \times 3} \to \mathbb{R}$ represents an oracle function evaluating the desired materials property. It can also be formulated as sampling from a tilted distribution $p(c, l, s) \exp(h(c, l, s))$ (Rafailov et al., 2024). Additional constraints can be integrated into the design process, allowing for flexible tasks such as compositional substitution (learning $p(c \mid l, s)$) and composition/structure completion (inpainting generation, learning $p(c^{\text{unknown}}, s^{\text{unknown}} \mid c^{\text{known}}, l, s^{\text{known}})$) (Dai et al., 2024).

### 2.2 (Meta)stability of Materials

Among computational approaches for evaluating crystal structure stability, Density Functional Theory (DFT) is the most reliable method for predicting formation energies in solid-state materials, showing close alignment with experimental measurements (Jain et al., 2011; Sun et al., 2016). The thermodynamic stability of a structure is quantified through its decomposition energy ($E_\text{d}$) with respect to the convex hull of known stable phases: $E_\text{d} = E_\text{s} - \sum_i x_i E_i$, where $E_\text{s}$ represents the total energy per atom, $x_i$ denotes the molar fraction of the $i$-th competing phase, and $E_i$ corresponds to its ground-state energy per atom. While the convex hull serves as a fixed reference, the evaluated structure $s$ need not be part of this hull. A negative decomposition energy ($E_\text{d} < 0$) indicates a thermodynamically stable state below the convex hull, while $E_\text{d} > 0$ suggests a metastable phase with a driving force for decomposition into more stable compounds. Our main objective for CSG is to identify stable crystal structures where $E_\text{d} \leq 0$.

Given the computational intensity of DFT calculations, universal Machine Learning Interatomic Potentials (MLIPs), trained on millions of DFT calculations, have emerged as efficient and reliable proxies for structure

stability assessment. Notable among these is CHGNet (Deng et al., 2023), a Graph Neural Network (GNN)-based MLIP that uniquely incorporates magnetic moments to capture both atomic and electronic interactions. M3GNet (Chen and Ong, 2022) offers an alternative approach, implementing three-body interactions in its graph architecture for accurate structural predictions across diverse chemical spaces. Recent advances in universal MLIPs include MACE (Batatia et al., 2023), DPA-1 (Zhang et al., 2024), and JMP (Shoghi et al., 2024), which demonstrate high accuracy in predicting crystal thermodynamic stability, particularly when trained on industrial-scale datasets comprising millions of compounds and non-equilibrium atomic configurations (Barroso-Luque et al., 2024; Merchant et al., 2023; Yang et al., 2024a). In this work, we employ the pre-trained CHGNet as our universal MLIP due to its closer alignment with DFT results, using a fixed phase diagram derived from the Materials Project 2023 DFT calculations (Jain et al., 2011; Wang et al., 2021).

## 3 MatLLMSearch

We propose MatLLMSearch, an evolutionary workflow that leverages pre-trained LLMs to search for stable and optimized crystal structures with. In this section, we introduce three key stages of the workflow as illustrated in Figure 1: (1) **Selection**, which identifies promising candidate structures from existing pools based on stability and property metrics; (2) **Reproduction**, where the LLM generates new candidates through implicit crossover and mutations of parent structures; and (3) **Evaluation**, which assesses proposed structures for validity, stability, and target properties. The overall workflow outlined in Algorithm 1 iteratively evolves a population of crystal structures while maintaining physical constraints and optimizing desired properties.

---

**Algorithm 1** The MatLLMSearch Framework

---

**Require:** Population size $K$, parent size $P$, reproduction size $C$, number of iterations $N$, known stable structures $\mathcal{D}$, oracle function $O$, extra pool $\mathcal{R}$

1:  ▷ *Initialization*
2:  Form population $\mathcal{P}_0$ by sampling $K$ groups of $P$ structures from $\mathcal{D}$
3:  Initialize structure collection $\mathcal{S} \leftarrow \varnothing$
4:  **for** $i \leftarrow 0, 1, \cdots, (N-1)$ **do**
5:      ▷ *LLM-guided reproduction*
6:      Generate prompts from parent structures in $\mathcal{P}_i$
7:      Obtain children structures $\mathcal{C}_i$ via LLM inference and parsing
8:      ▷ *Structure evaluation*
9:      Relax structures $\mathcal{C}_i \leftarrow \text{CHGNetRelax}(\mathcal{C}_i)$
10:     Calculate decomposition energy $E_\text{d}$ and properties
11:     Evaluate objective scores using oracle function $O$
12:     Update structure collection $\mathcal{S} \leftarrow \mathcal{S} \cup \mathcal{C}_i$
13:     ▷ *Selection*
14:     Form candidate pool from parents $\mathcal{P}_i$, children $\mathcal{C}_i$, and extra pool $\mathcal{R}$
15:     Select top-$(K \times P)$ structures based on objective scores from the candidate pool
16:     Construct next parent groups $\mathcal{P}_{i+1}$
17: Validate final structures via DFT
18: **return** cumulated structures $\mathcal{S}$

---

### 3.1 Initialization

We first optionally sample an **extra pool** $\mathcal{R}$ of reference structures from a database of known stable structures $\mathcal{D}$. $\mathcal{R}$ is used to initialize the population and will be considered for fitting during following iterations. Our evolutionary search begins by sampling $(K \times P)$ structures from $\mathcal{D}$ to form the initial parent pairs $\mathcal{P}_0$. These structures are randomly paired into $K$ groups of $P$ parents each to serve as reference examples in LLM prompts. $\mathcal{R}$ can be customized to suit various design objectives, with more details and ablation studies provided in Section 4.4.

## 3.2 Reproduction

Genetic algorithms traditionally mimic biological evolution through explicit crossover and mutation operations (Heiles and Johnston, 2013; Johnston, 2003). In CSP, crossover typically involves combining structural fragments from parent structures (e.g., swapping atomic positions or structural motifs), while mutation introduces random variations through predefined operations like atomic displacement, lattice transformation, or element substitution (Curtis et al., 2018; Kadan et al., 2023). While effective, these rigid operators can limit the exploration of the complex crystal structure space. In MatLLMSearch, we explore the flexibility of LLMs for structure reproduction. Through prompt-based guidance, we ask LLMs to perform implicit crossover and mutation by analyzing and combining structural information from parent materials. Specifically, LLMs are instructed to "modify or combine the base materials", while maintaining chemical validity and enhancing target properties. This approach allows LLMs to freely and simultaneously introduce variations across multiple structural aspects, including atomic positions, lattice parameters, and element substitutions, or even generate completely new structures functionally relevant to parent structures.

## 3.3 Evaluation

Our evaluation pipeline consists of two stages:

- **Rule-based validation** ensures structural integrity by verifying three-dimensional periodicity, physical connectivity (interatomic distances between 0.6–1.3 times the sum of atomic radii), and chemical validity through charge balance analysis.

- **Stability and property evaluation** begins with structure relaxation using CHGNet. We quantify thermodynamic stability through decomposition energy $E_d$ calculated as the distance to the Materials Project convex hull. Notably, we observe that LLM-proposed structures typically require minimal relaxation, with 61.1% of structures exhibiting small energy changes ($|\Delta E| < 0.5$ eV/atom) during this process (detailed in Appendix G.2), indicating their initial stability. For stability-focused optimization, we quantify thermodynamic stability through the decomposition energy $E_d$ using CHGNet, calculated as the distance to the convex hull from the Materials Project database (version 2023-02-07-ppd-mp).

  For multi-objective optimization, additional properties such as bulk modulus can be evaluated. These quantitative scores then guide the selection process for subsequent generations, allowing our framework to flexibly adapt to different design goals.

## 3.4 Selection

Last, the selection stage evolves a population of candidate structures that meet the optimization objectives, such as thermodynamic stability or other desired physical properties. For each iteration $i$, we construct a new parent pool $\mathcal{P}_{i+1}$ of the same size ($K \times P$) by selecting top-ranked candidates from three sources: (a) the current parent pool ($\mathcal{P}_i$), (b) newly generated children structures ($\mathcal{C}_i$), and (c) an optional extra pool ($\mathcal{R}$) to improve diversity. Candidates in $\mathcal{P}_i \cup \mathcal{C}_i \cup \mathcal{R}$ are ranked according to optimization objectives, either single-objective (e.g., $E_d$ for stability) or multi-objective criteria (e.g., alternating between different properties).

## 3.5 Final DFT Verification

After completing all evolutionary iterations, we collect the cumulated children structures $\mathcal{S} = \bigcup_i \mathcal{C}_i$ for final validation using DFT. To save computational cost, we focus on meta-stable structures with CHGNet-predicted decomposition energy $E_d < 0.1$ eV/atom. DFT calculations are performed using VASP 6 in the Generalized Gradient Approximation (GGA) with PBE functional (Perdew et al., 1996), using the projector-augmented wave method (Kresse and Furthmüller, 1996; Kresse and Joubert, 1999). We employed a plane-wave basis set with an energy cutoff of 520 eV and a $k$-point mesh of 1,000 per reciprocal atom (Jain et al., 2013). The calculations converged to $10^{-6}$ eV in total energy for electronic self-consistent field cycles and 0.02 eV/Å in

interatomic forces for the ionic steps. The computational settings are consistent with MPGGARelaxSet and MPGGAStaticSet (Jain et al., 2011).

## 4 Experiments

### 4.1 Experimental Settings

For the main experiments, we use Llama 3.1 (70B) (Grattafiori et al., 2024) with temperature 0.95 as the base LLM. The evolution performed with parent size $P = 2$, reproduction size $C = 5$, and population size $K = 100$ for $N = 10$ iterations unless otherwise specified. Crystal structures are represented in the POSCAR format with 12 decimal digits.

For initialization, we use the MatBench-bandgap dataset (Dunn et al., 2020) as the reference database $\mathcal{D}$. We sample some known stable structures from $\mathcal{D}$ to form the initial generation; they serve as the candidates for parents during each iteration. Samples are selected based on minimizing CHGNet-predicted decomposition energy or optimizing specific target properties. Detailed hyper-parameter sensitivity analysis is provided in Appendix G.

### 4.2 Evaluation Metrics

We evaluate our framework on three primary aspects: (a) stability, (b) validity and diversity, and (c) novelty. We would like to emphasize that stability is the most important metric among all to reflect generation quality. Additionally, we extend the evaluation to computational efficiency and provide detailed definitions for all metrics in Appendix D.

**Stability.** Thermodynamic stability is the primary criterion for material realizability. We quantify this via the decomposition energy ($E_\mathrm{d}$) relative to the Materials Project convex hull (version `2023-02-07-ppd-mp`). The gold standard for stability is Density Functional Theory (DFT) verification, where a structure is stable only if its DFT-calculated $E_\mathrm{d} \leq 0.0$ eV/atom. Structures identified as metastable ($E_\mathrm{d} < 0.1$ eV/atom) by CHGNet undergo further DFT calculations for stability assessment. For a fair comparison with baselines, we also report metastability rates evaluated with multiple MLIPs (CHGNet, M3GNet, and Orb-v3 for CSP). We note that the DFT verification should be the primary validation metric. We use multi-MLIP metastability rates for comparison with baselines and they are not substitutes for first-principles DFT calculations.

**Diversity.** To evaluate the diversity of our generated structures, we analyze the specific characteristics including element co-occurrence pattern and space group distribution, comparing LLM-generated structures with reference structures from MatBench-bandgap that forms the initial generation. In addition, we analyze the compositional and structural diversity among generated structures. The discussion is detailed in Appendix H.

**Novelty.** A structure is compositionally novel if its reduced formula is absent from the reference database (MatBench-bandgap, $\sim 106$K structures). It is structurally novel if no match is found using `StructureMatcher` of `pymatgen` with default settings.

**S.U.N. rate.** Following the protocol of FlowMM (Miller et al., 2024) and MatterGen (Zeni et al., 2025), we report the S.U.N. (Stable, Unique, Novel) rate as a supplementary generation quality metric. We calculate it as the ratio of generated structures that are simultaneously stable (DFT $E_\mathrm{d} \leq 0.0$ eV/atom after CHGNet pre-relaxation), unique (no match in the generated set via `StructureMatcher`), and novel (no match in the reference database via `StructureMatcher`). We emphasize that direct cross-method comparison is limited by the reference sets used for novelty calculation. As illustrated in Figure 4, while baselines evaluate novelty against their training sets, we compare against the full MatBench-bandgap dataset.

### 4.3 Main Experimental Results

In this section, we evaluate our proposed pipeline on progressively more challenging tasks, from crystal structure generation through design to crystal structure prediction.

| Model | $f$-ele in Parents[†] | Validity | | Metastability | | | Stability[‡] | |
| --- | --- | --- | --- | --- | --- | --- | --- | --- |
| | | | | M3GNet | CHGNet | | DFT | |
| | | Structural | Composition | $E_\mathrm{d} < 0.1$ | $E_\mathrm{d} < 0.1$ | $E_\mathrm{d} < 0.03$ | w/ $f$-ele | w/o $f$-ele[§] |
| CDVAE* | — | 100.0% | 86.7% | 28.8% | — | — | 5.4% | — |
| CrystalTextLLM-7B* | — | 96.4% | 93.3% | 35.0% | — | — | 8.7% | — |
| CrystalTextLLM-13B* | — | 95.5% | 92.4% | 38.0% | — | — | 14.4% | — |
| CrystalTextLLM-70B* | — | 99.6% | 95.4% | 49.8% | — | — | 10.6% | — |
| MatLLMSearch | ✓ | 100.0% | 85.6% | 81.1% | 76.8% | 56.5% | 31.7% | 14.0% |
| (Llama 3.1-70B) | ✗ | 100.0% | 89.0% | 81.9% | 78.4% | 54.8% | 27.0% | 24.6% |

Table 1: Performance comparison of crystal structure generation. Metastability is first assessed using surrogate models, where we report both M3GNet and CHGNet results for fair comparison with baselines CDVAE and CrystalTextLLM (which use M3GNet). *Results taken from the original papers. [†]Indicates whether $f$-electron elements are excluded in parent structures (not applicable to CDVAE and CrystalTextLLM as they are trained on data including $f$-electron elements). [‡]The stable fraction represents the percentage of DFT-verified stable structures ($E_\mathrm{d} < 0.0$ eV/atom) over structures predicted to be metastable ($E_\mathrm{d} < 0.1$ eV/atom) by respective surrogate models (M3GNet for CDVAE and CrystalTextLLM, CHGNet for ours, with CHGNet being more rigorous as evidenced by lower metastability rates). [§]We exclude structures containing $f$-electron in DFT verification while keeping the denominator as all metastable structures.

### 4.3.1 Crystal Structure Generation

**Stability.** We first evaluate the ability of our framework to generate stable crystal structures by optimizing decomposition energy $E_\mathrm{d}$ as the sole objective. The LLM prompting template is provided in Appendix C.

We compare our model against two baseline models CDVAE (Xie et al., 2022) and CrystalTextLLM (Gruver et al., 2024) in Table 1. Following baselines, we report structural and compositional validity as simple sanity check, which assess non-overlapping atomic radii and charge neutrality respectively. LLM output passes the simple sanity check at a high validity rate. Then, we evaluate the 1,479 generated structures for metastability with MLIPs, where 76.8% and 81.1% are metastable based on CHGNet and M3GNet evaluations respectively. Performance evaluated by both MLIPs outperform the 49.8% metastability rate by M3GNet of the state-of-the-art CrystalTextLLM 70B model, which has a comparable model size to our base model. We then perform DFT validation for stability assessment, 31.7% of the metastable structures remain stable, substantially improving the 10.6% stability rate from CrystalTextLLM 70B.

However, structures containing $f$-electron elements (actinides and lanthanides, abbreviated as $f$-ele) challenge stability prediction with their strongly correlated electron interactions, which may not be adequately captured by DFT approaches under GGA and Hubbard $U$ corrections (Himmetoglu et al., 2014). These structures consistently yield lower decomposition energies ($E_\mathrm{d}$), creating a potential computational shortcut. To assess this effect, we report the percentage of stable structures without $f$-ele (denoted as "w/o $f$-ele") among the metastable structures. By excluding $f$-electron structures from parent selection (marked with ✗), we improved metastability rates to 78.4% and increased stable non-$f$-electron structures from 14.0% to 24.6%. This simple intervention demonstrates the ability of our framework to effectively guide exploration toward diverse stable configurations, which remain unaddressed by existing methods.

**Diversity.** To evaluate the diversity of our generated structures, we analyzed their compositional and structural characteristics by comparing LLM-proposed structures and with the $(K \times P)$ most metastable structures from MatBench that forms the initial generation. Our element co-occurrence analysis reveals high compositional diversity in the LLM-proposed structures, with even the most frequent compositions appearing only twice (approximately 0.14% of total structures). Examination of element co-occurrences with F in Figure 2 highlights the effectiveness of our evolutionary method in guiding structure generation toward more stable F-based compounds, particularly with alkali metals and transition metals.

The structural diversity is further evidenced by space group distribution and stability analysis in Figure 7(b). The generated structures demonstrate broad structural diversity with high metastability rates across multiple space groups, confirming that our evolutionary method navigates toward stable regions of chemical space

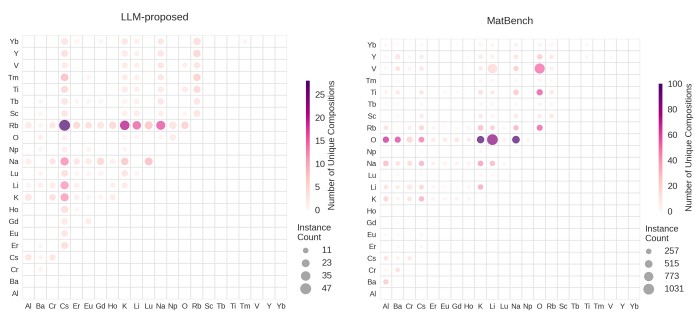

Figure 2: Element co-occurrence patterns with fluorine (F) in LLM-proposed structures (left) versus MatBench structures (right). Bubble size indicates frequency of occurrence for each element pair, while color intensity represents compositional diversity (darker indicates more unique compositions with that element pair).

Figure 3: Decomposition energy ($E_d$) distribution comparison across experimental configurations. Vertical lines indicate metastable thresholds at 0.0 eV/atom and 0.1 eV/atom. Reference-guided approaches show more balanced distributions.

while maintaining diverse structural motifs across different crystallographic symmetries. Additional diversity and novelty evaluations and analysis are provided in Appendix H.

| Method | Metastability ($E_d < 0.0$) | Stability | S.U.N. |
|---|---|---|---|
| MatLLMSearch (Llama 3.1-70B) | 37.59% | 24.34% | 21.64% |
| DiffCSP | — | 5.06% | 3.34% |
| FlowMM | — | 4.65% | 2.34% |

Table 2: (Meta)stability and S.U.N. rates comparison. S.U.N. computed against the entire MatBench-bandgap dataset.

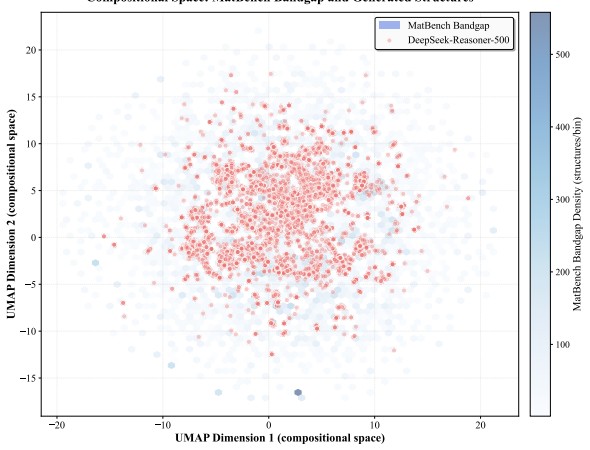

Figure 5: Compositional space overlap (UMAP projection).

Figure 4: Difference in comparing the S.U.N. rate in different baselines.

**S.U.N. rate.** The results in Table 2 demonstrate that our framework maintains higher stability and S.U.N. (Stable, Unique, Novel) rates compared to the two baselines. Specifically, we achieve a 37.59% metastable rate ($E_d < 0.0$ eV/atom) with CHGNet, a 24.34% stable rate verified by DFT calculations, and an overall 21.64% S.U.N. rate, outperforming both DiffCSP (Jiao et al., 2024) and FlowMM (Miller et al., 2024).

However, we caution that the S.U.N. metric has inherent limitations: it is dependent on the chosen reference sets and collapses complex, multi-dimensional properties into binary counts. Figure 4 illustrates the key methodological difference in S.U.N. computation: while FlowMM and DiffCSP evaluate novelty against their training sets, our framework compares against the entire MatBench dataset (106,113 structures). Consequently, we treat S.U.N. primarily as a supplementary signal for novelty assessment rather than a definitive metric.

To strengthen the analysis, we also provide comprehensive evaluation covering all S.U.N. dimensions in Appendix H. This includes DFT-calculated stability; metastability verficiation with multiple MLIPs (CHGNet, M3GNet and Orb-v3 for CSP); space group and crystal system distributions for Uniqueness; and compositional

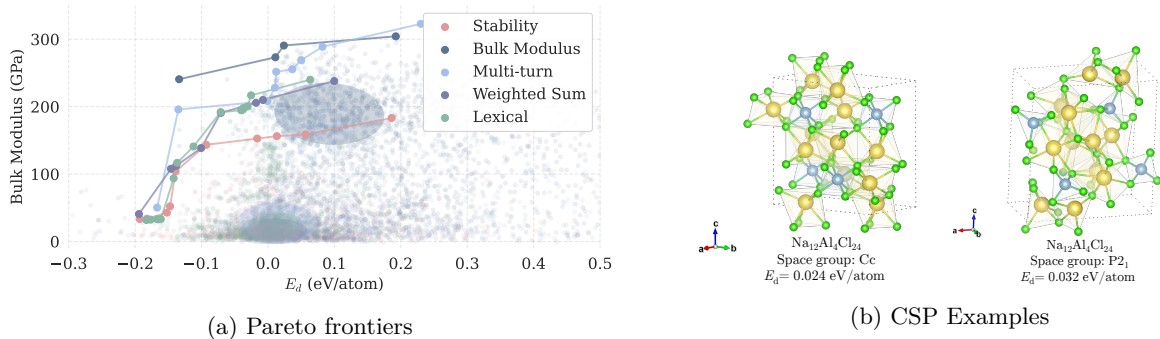

(a) Pareto frontiers

(b) CSP Examples

Figure 6: (a) Pareto frontiers of bulk modulus versus decomposition energy ($E_{\mathrm{d}}$) for structures optimized towards stability, bulk modulus and multi-objective (multi-turn). Ellipses indicate regions of highest structure density. (b) Examples of predicted crystal structures with composition $Na_3AlCl_6$.

and structural novelties, elemental co-occurrence pattern shifts for Novelty. Further discussion on the fairness of these baseline comparisons is available in Appendix D.

**Extended evaluation.** To better compare with baseline methods, we perform a larger scale CSG experiment with DeepSeek-Reasoner, with experimental settings detailed in Appendix G.5. Figure 5 shows the compositional space overlap of generated structures and MatBench reference structures using UMAP projection of Magpie features. The generated structures mostly overlap with the MatBench compositional space. Additional hyper-parameter sensitivity analysis including model scaling effects, fine-tuning comparisons, reproduction parameters, representation format, base LLM choices, etc. are detailed in Appendix G. Besides performance gains, our method also demonstrates computational advantages. A detailed efficiency analysis, including generation time overhead and carbon footprint, is provided in Appendix G.6.

### 4.3.2 Crystal Structure Design

We also explore multi-objective optimization by extending our framework to balance stability with desired material properties. We demonstrate this capability by alternating between optimizing stability ($E_{\mathrm{d}}$) and bulk modulus in each iteration. While this multi-objective setting naturally yields lower stability rates (57.1% metastable with $E_{\mathrm{d}} < 0.1$ eV/atom and 15.6% DFT-verified stable structures with $f$-electron elements) compared to stability-only optimization, it enables the discovery of structures with favorable property-stability trade-offs.

As shown in Figure 6(a), the Pareto frontiers under various optimization strategies converge in regions with high bulk modulus ($> 200$ GPa) and metastability ($E_{\mathrm{d}} \leq 0.1$ eV/atom) in the stability-property space, indicating successful discovery of potentially valuable structures that balance both objectives. The regions of highest structure density, estimated using Gaussian KDE and visualized as ellipses, reveal how optimization goals affect the distribution. Prioritizing bulk modulus shifts the density distribution toward higher mechanical strength at the cost of increased decomposition energy. We provide additional discussions of multi-objective optimization strategies in Appendix E.

### 4.3.3 Crystal Structure Prediction

We next evaluate our framework on crystal structure prediction tasks, which aim to predict stable structure (i.e. lattice and atomic coordinates) for a given composition. As a case study, we prompt the LLM to predict polymorphs of $Na_3AlCl_6$. For context, the Materials Project database currently contains only one structure for this composition (`mp-1111450`, $Fm\bar{3}m$, $E_{\mathrm{d}} = 0.142$ eV/atom), which is significantly unstable.

During the prompting process, we apply specific structural filters to select seed structures containing only three distinct elements in a 3:1:6 ratio, matching the stoichiometry of $Na_3AlCl_6$. From MatBench, we identified 820 structures meeting the criteria to build the initial population. Example structures proposed by the LLM for this composition are visualized in Figure 6(b), with DFT-verified decomposition energies

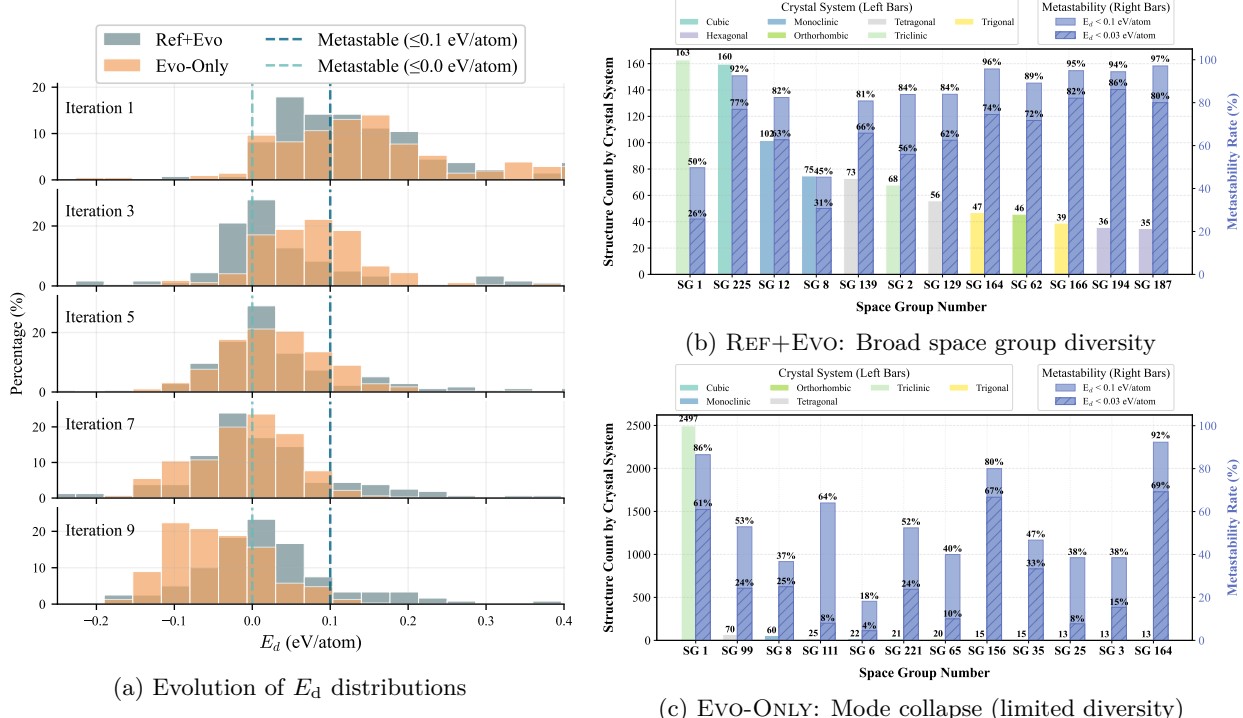

(a) Evolution of $E_d$ distributions

(b) REF+EVO: Broad space group diversity

(c) EVO-ONLY: Mode collapse (limited diversity)

Figure 7: Ablation analysis comparing reference-guided (REF+EVO) vs. reference-free (EVO-ONLY) generation. (a) Iterative optimization shifts $E_d$ distributions toward metastability. (b) The reference-guided REF+EVO maintains diverse crystallographic symmetries. (c) Without references, EVO-ONLY suffers from diversity collapse, concentrating on limited space groups.

of 0.024 and 0.032 eV/atom respectively. Although these predicted polymorphs remain metastable, their decomposition energies $E_d$ are significantly lower than the previously reported structure in MatBench ($E_d$ reduced by up to 83%), exemplifying the potential of our evolutionary pipeline for CSP applications. For readers of interest, we demonstrate additional successful structure prediction cases including $Ag_6O_2$, $Bi_2F_8$, etc. in Appendix F.

### 4.4 Ablation Analysis

We evaluate four configurations to disentangle the impact of reference structures and evolutionary iterations: REF+EVO (the proposed framework), EVO-ONLY (evolutionary search without references), REF-1SHOT (single-iteration generation with references), and ZERO-SHOT (standard zero-shot generation).

**Thermodynamic stability analysis.** The decomposition energy distributions in Figure 3 highlight the critical role of the components of MatLLMSearch. The reference-guided approaches (REF+EVO, REF-1SHOT) effectively concentrate structures near the metastable threshold ($E_d \approx 0.0$ eV/atom), indicating high thermodynamic quality. While evolutionary search alone (EVO-ONLY) achieves a similar median stability, its distribution is more dispersed. In contrast, the ZERO-SHOT baseline yields structures with substantially inferior thermodynamic stability, confirming that neither references nor evolution can be entirely omitted without performance loss.

**Impact of evolutionary iterations.** Evolutionary iterations are critical for maximizing generative yield and thermodynamic stability. As shown in the decomposition energy distributions (Figure 7(a)), the $E_d$ distribution progressively shifts toward greater stability over successive iterations. Further, under an equivalent computational budget of 1,000 LLM inferences, REF+EVO yields 1,479 valid structures compared to only 741 for REF-1SHOT. This confirms that iterative refinement not only optimizes stability but also improves the acceptance rate of generated candidates.

**Impact of reference structures.** References act as critical geometric priors that prevent mode collapse. While evolutionary search alone (Evo-Only) successfully optimizes metastability, it suffers from severe mode collapse. Figure 7(c) reveals that 88% of its generated structures collapse into the triclinic space group 1, indicating convergence to a repetitive crystallographic motif. Conversely, the reference-guided model Ref+Evo (Figure 7(b)) maintains high metastability while preserving a broad distribution across diverse space groups. This suggests that reference structures provide necessary geometric diversity, preventing the optimization process from collapse into repetitive solutions.

**Summary.** Our ablation study reveals distinct trade-offs: Evo-Only optimizes stability but suffers from diversity collapse. Ref-1Shot ensures diversity but yields limited quantity. Zero-Shot fails to achieve sufficient quality or quantity. By synergizing reference structures with evolutionary search, Ref+Evo simultaneously achieves high metastability, balanced structural diversity, and scalable generative yield.

# 5 Related Work

## 5.1 Language Models for Materials Science

The increasing capabilities of LLMs have prompted materials science community to explore their potential for understanding and predicting material properties (Jablonka et al., 2023). Benchmarking studies (Rubungo et al., 2024) also suggest that fine-tuning LLMs over specific materials datasets can lead to comparable or better performance than specialized graph neural networks.

Recent autoregressive approaches has developed along two main paths. First is to train from scratch. Flam-Shepherd and Aspuru-Guzik (2023) demonstrate that autoregressive models trained with character-level tokenization can generate chemically valid crystal structures by directly tokenizing CIF files into string sequences. Other approaches include Wyckoff Transformer (Kazeev et al., 2025) for symmetric crystal generation, deCIFer (Johansen et al., 2025) for crystal structure prediction from powder diffraction data, and multimodal foundation models (Moro et al., 2025) for material property prediction. Secondly, CrystalTextLLM (Gruver et al., 2024) fine-tunes a pre-trained LLM (over massive texts) on generating crystalline structures with task-specific prompts. Mat2Seq (Yan et al., 2024) converts 3D crystal structures into unique 1D sequences that preserve $SE(3)$ and periodic invariance for language model training. While these approaches produce valid structures, they sacrifice the general conversation capabilities of LLMs due to specialized training or fine-tuning on crystallographic data. Recent work also raised concerns about the scalability and reliability of LLMs for real-world materials discovery (Miret and Krishnan, 2024).

In parallel developments within molecular chemistry, MolLEO (Wang et al., 2025) successfully employs pre-trained LLMs without domain-specific fine-tuning to search for small molecules. Subsequent work (Lu et al., 2024) extended this evolutionary optimization approach to more complex transition metal chemistry using advanced base LLMs with enhanced reasoning capabilities. However, these applications benefit from natural string representations for molecules (e.g., SMILES or SELFIES), which are considerably simpler than the three-dimensional representations required for crystal structures. Our work bridges this gap by adapting the evolutionary approach to the more complex domain of crystal structures without requiring fine-tuning.

## 5.2 Generative Models for Materials Discovery

Besides autoregressive language models, various generative models including variational autoencoders, diffusion models, and flow models have emerged as promising solutions for crystal structure generation. Early work proposes generative crystal structures using variational autoencoders that represent crystal structures as 3D voxels (Court et al., 2020; Noh et al., 2019). CDVAE first proposes to generate crystal structures with a score-based generative (diffusion) model and optimize crystal structure properties through gradient-based optimization in the latent space (Xie et al., 2022). This approach has been extended in several directions: Jiao et al. (2024) developed Riemannian diffusion models to better handle periodic coordinates, Zeni et al. (2025) scaled the approach to encompass elements across the entire periodic table with various design criteria, and Dai et al. (2024) applied it to crystal inpainting tasks. Most recently, Miller et al. (2024); Sriram et al. (2024) introduced Riemannian flow matching models to better address periodic boundary conditions with improved

| Method | Category | Genetic Operators | Key Innovation |
|---|---|---|---|
| USPEX (Lyakhov et al., 2013; Oganov and Glass, 2006) | Classical EA | Symmetric crossover, lattice mutation | Pioneering evolutionary CSP with heredity operators |
| XtalOpt (Falls et al., 2020; Lonie and Zurek, 2011) | Classical EA | Multiple crossover schemes | Flexible operator combinations |
| GASP (Curtis et al., 2018) | Classical EA | Crossover, mutation, permutation | Python framework for alloys |
| CALYPSO (Wang et al., 2012) | Classical EA | PSO-inspired operators | Particle swarm integration |
| MAISE (Balachandran et al., 2016) | ML-Enhanced | Classical + ML screening | 10–100× speedup via active learning |
| MAGUS (Wang et al., 2023) | ML-Enhanced | Graph-decomposed operators | On-the-fly ML + graph theory |
| GOFEE (Jennings et al., 2019) | Bayesian Opt | Gaussian process-guided | Surrogate-based convergence |
| CrySPY (Yamashita et al., 2021) | Bayesian Opt | Modular EA + BO | Flexible multi-method framework |
| **MatLLMSearch** | **LLM-Guided** | **Implicit mutation and crossover in LLM generation** | **EA + LLM for CSG, CSP, and CSD** |

Table 3: Comparison of evolutionary CSP methods.

performance. Space group-aware methods have further advanced this direction: CrystalFormer (Cao et al., 2025) uses space group informed transformers, while Chang et al. (2025); Puny et al. (2025) incorporate space group equivariance and conditional flow matching, respectively. Universal models for atoms (Wood et al., 2025) provide a unified framework for materials modeling. Yang et al. (2024b) explore the synergy between language and generative models by leveraging LLMs to propose chemical formulae under design constraints before feeding them to a diffusion model.

### 5.3 Evolutionary Algorithms in Materials Discovery

Evolutionary algorithm has also played an important role in materials discovery, especially Crystal Structure Prediction (CSP), where the goal is to identify the global energy minimum for a given chemical composition (Oganov and Glass, 2006; Wang et al., 2012). Representative methods include USPEX (Lyakhov et al., 2013; Oganov and Glass, 2006), XtalOpt (Falls et al., 2020; Lonie and Zurek, 2011), GASP (Curtis et al., 2018), CALYPSO (Wang et al., 2012), and GAmuza (Kadan et al., 2023), which employ hand-crafted genetic operators such as symmetric crossover by swapping atomic positions while preserving one parent's space group and lattice mutation by perturbing crystallographic parameters. These methods have predicted stable phases for systems including gamma-boron (Oganov et al., 2009), high-pressure sodium compounds (Ma et al., 2009), and metal hydrides (Peng et al., 2017). Recent approaches integrate machine learning to accelerate CSP, including active learning by MAISE (Balachandran et al., 2016), graph-based structure decomposition by MAGUS (Wang et al., 2023), and Bayesian optimization methods like GOFEE (Jennings et al., 2019) and CrySPY (Yamashita et al., 2018; 2021).

We summarize key characteristics of established evolutionary CSP methods in Section 5.3. Recent benchmarks (Duval et al., 2025) and reviews (Handoko and Made, 2025) provide broader context on the landscape of generative models for materials discovery. DiffCSP (Jiao et al., 2024) achieves higher match rate (73.33% over 53.33%) and much faster inference (10 seconds over 12.5 hours) than USPEX when evaluated on 10 binary and 5 ternary compounds in the MP-20 test set.

## 6 Conclusion

In this paper, we present an evolutionary workflow for computational materials discovery, encompassing crystal structure generation, prediction, and objective-based optimization. We demonstrate that a pre-trained LLM trained on general text can identify a higher proportion of (meta)stable materials compared to state-of-the-art generative models specifically trained on materials datasets. These findings suggest that LLMs inherently function as effective crystal structure generators, with both compositional and structural information naturally embedded in their text-inference capabilities. In conclusion, our method complements existing structure discovery techniques by providing refined optimization capabilities while maintaining versatility in addressing various optimization objectives. The stability of generated materials is related to both structural geometry and chemical composition (Szymanski and Bartel, 2025). Our LLM-based evolutionary approach achieves

superior performance by populating the energy-favorable chemical compositions using an MLFF-based oracle function that provides strong guidance unavailable in other *de novo* crystal generative models.

Looking forward, a natural extension of this work would be synthesis prediction based on the evolutionary method. Improved machine learning interatomic potentials will complement this process, as discussed in Appendix D. Such development would benefit from integration with high-quality experimental data from automated, high-throughput experiments, bridging the gap between computational predictions and experimental synthesis, which would accelerate high-throughput materials discovery.

**Limitations and broader impact.** This work aims to advance machine learning and computational materials discovery by making crystal structure generation more accessible and efficient. This advancement will particularly benefit researchers who have limited computational resources, enabling scientific discovery without the need to train large ML models. Additionally, the oracle functions can be further adapted to incorporate experimental data or high-fidelity property predictors for the generated crystal structures when applying this pipeline to practical materials discovery.

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

# Supplementary Material for **MatLLMSearch**

# A    Experimental Details and Reproducibility

The extra pool $\mathcal{R}$ of stable structures is sampled from the MatBench-bandgap dataset (Dunn et al., 2020), which consists of 106,113 crystal structures in total. To initialize the parent structures for the first iteration, we rank candidate structures by decomposition energy computed with CHGNet and then apply de-duplication by composition. For the CSG task, we remove binary compounds and structures with higher-order compositions, retaining candidates with 3 to 6 elements. For CSP tasks, we filter seed structures to match a desired compositional pattern and also apply de-duplication by composition.

The implementation of our evolutionary search pipeline is available here. The crystal structures generated by MatLLMSearch that are presented in the main results can be downloaded here. We provide structures parsed directly from LLM responses, as well as structures after CHGNet relaxation. We also provide the generated structures and reference structures for the larger scale experiment conducted with DeepSeek-Reasoner.

# B    Machine Learning Interatomic Potentials

A significant breakthrough in addressing computational cost challenges has emerged through the development of machine learning interatomic potentials (MLIPs) trained on high-fidelity quantum mechanical calculations (e.g., DFT) (Batzner et al., 2022; Cheng, 2024; Du et al., 2023a;b; Liao et al., 2024; L'opez-Zorrilla et al., 2023; Yin et al., 2025; Zhang et al., 2021). In MLIPs, the total energy is expressed as a sum of atomic contributions, where each atom's energy depends on its local environment including the atomic coordinates and chemical species of neighboring atoms within a cutoff radius:

$$\hat{E} = \sum_{i}^{n} \phi(\{\vec{r}_j\}_i, \{C_j\}_i), \quad \hat{\boldsymbol{f}}_i = -\frac{\partial \hat{E}}{\partial \boldsymbol{r}_i}, \quad \boldsymbol{\sigma} = \frac{1}{V}\frac{\partial \hat{E}}{\partial \boldsymbol{\varepsilon}}. \tag{S1}$$

Here, $\phi$ is a learnable function that maps the set of position vectors $\{\vec{r}_j\}_i$ and chemical species $\{C_j\}_i$ of the neighboring atoms $j$ to the energy contribution of atom $i$. The forces $\boldsymbol{f}_i$ and stress $\boldsymbol{\sigma}$ are calculated via auto-differentiation of the total energy with respect to the atomic Cartesian coordinates and strain. Recent advances have demonstrated that MLIPs, trained on extensive density functional theory (DFT) calculations accumulated over the past decade across diverse materials systems, exhibit remarkable transferability in performing atomistic simulations across various material and chemical systems. These broadly applicable potentials are known as universal MLIPs (uMLIPs) (Batatia et al., 2023; Chen and Ong, 2022; Deng et al., 2023; Park et al., 2024). By leveraging uMLIPs as surrogate energy models, researchers can rapidly optimize crystal structures and obtain structure-energy relationships for assessing thermodynamic stability. Recent benchmark studies, including MACE (Batatia et al., 2023), DPA-1 (Zhang et al., 2024) and JMP (joint multi-domain pretraining) (Shoghi et al., 2024), have demonstrated the high accuracy of these uMLIPs in predicting crystal thermodynamical stability, particularly for industrial-scale implementations trained on millions of compounds and non-equilibrium atomic configurations (Barroso-Luque et al., 2024; Merchant et al., 2023; Yang et al., 2024a).

To accelerate the oracle function evaluation in the evolutionary iterations, we performed all structure relaxations with the FIRE optimizer (Bitzek et al., 2006) over the potential energy surface provided by CHGNet, where the atom positions, cell shape, and cell volume were optimized to reach converged interatomic forces of 0.1 eV/atom (Deng et al., 2023). The output energy prediction is directly compatible with the Materials Project phase diagrams with the MaterialsProject2020Compatibility (Wang et al., 2021).

## C   Prompt Templates

*You are an expert material scientist. Your task is to propose hypotheses for {reproduction_size} new materials with valid stable structures and compositions. No isolated or overlapped atoms are allowed.*

*The proposed new materials can be a modification or combination of the base materials given below.*

*Format requirements:*

    *1. Each proposed structure must be formatted in JSON with the following structure:*

```
{{
    "i":  {{
        "formula":  "composition_formula",
        "POSCAR":  "POSCAR_format_string"
    }}
}}
```

    *2. Use proper JSON escaping for newlines (\n) and other special characters*

*Base material structure for reference:*
    *{reference_structures}*

*Your task:*
    *1. Generate {reproduction_size} new structure hypotheses*
    *2. Each structure should be stable and physically reasonable*
    *3. Format each structure exactly as shown in the input*

*Output your hypotheses below:*

## D   Details on Evaluation Metrics

### D.1   Stability

Thermodynamic stability is the primary criterion for material realizability. We quantify this using the decomposition energy ($E_\mathrm{d}$), which measures the energy distance of a structure to the convex hull of known stable phases.

**Reference hull.** All decomposition energies are computed against the Materials Project phase diagram (version `2023-02-07-ppd-mp`).

**Metastability.** For high-throughput evaluation, we use Machine Learning Interatomic Potentials (MLIPs) including CHGNet, M3GNet, and Orb-v3. A structure is defined as metastable if its predicted $E_\mathrm{d}$ falls below a chosen threshold (e.g., 0.1 eV/atom, 0.03 eV/atom, or 0.0 eV/atom).

**Stability.** The gold standard for stability is verification via Density Functional Theory (DFT). A structure is classified as *stable* only if its DFT-calculated $E_\mathrm{d} \leq 0.0$ eV/atom.

### D.2   Diversity

We extend the evaluation on structural and compositional diversity using established metrics from prior work (Gruver et al., 2024; Xie et al., 2022).

**Compositional diversity and structural diversity.** For each generated structure, we compute a composition fingerprint and a structural fingerprint using featurizers from the `Matminer` library. Then, we calculate the mean pairwise cosine distances between fingerprints of the generated structures. Log normalization is applied to composition diversity for 0-1 scale standardization.

## D.3 Novelty

**Reference database.** We evaluate novelty against the entire **MatBench-bandgap** dataset ($\sim$106K structures), which is the source of our parent pools.

**Compositional novelty.** A structure is compositionally novel if its `reduced_formula` does not exist in the reference database.

**Structural novelty.** A structure is considered structurally novel if it has no match in the reference database using `StructureMatcher` from `pymatgen` with default settings: `ltol=0.2`, `stol=0.3` Å, `angle_tol=5°`.

## D.4 S.U.N. Rate

Following the protocol of FlowMM (Miller et al., 2024) and MatterGen (Zeni et al., 2025), we report the S.U.N. (Stable, Unique, Novel) rate as a supplementary generation quality metric. We compute S.U.N. as the fraction of generated structures that are simultaneously: *stable* (DFT-verified, $E_d \leq 0.0$ eV/atom after CHGNet pre-relaxation), *unique* (no match within the generated set via `StructureMatcher`), and *novel* (no match in the reference database via `StructureMatcher`).

**Limitations for cross-method comparison.** S.U.N. collapses multiple axes of quality into a single number and is sensitive to the reference set used for novelty. This dependence complicates direct cross-method comparison, because baseline methods typically evaluate novelty against their training sets, whereas our framework compares against the full MatBench-bandgap dataset ($\sim$106K structures). This methodological discrepancy is detailed in Section 4.2 and illustrated in Figure 4. We therefore use S.U.N. as a supplementary signal, while treating DFT verification as the primary stability criterion.

For the main results reported in the text, "Stable" in S.U.N. refers to DFT verification ($E_d \leq 0.0$ eV/atom). For the extended hyper-parameter analysis across multiple base LLMs in Table S9, we adopt a CHGNet-based criterion ($E_d < 0.0$ eV/atom) due to the computational cost of DFT.

To provide a comprehensive view of generation quality beyond S.U.N., we also report fine-grained evaluations of each dimension. For stability, we report both DFT-verified stability rates and metastability rates computed with multiple MLIPs (CHGNet and M3GNet for CSG; Orb-v3 for CSP). For uniqueness, we analyze space group distributions, crystal system diversity, and compositional space coverage. For novelty, we report compositional and structural novelty in Figure S3 and show elemental co-occurrence patterns in Figure 2 to evidence exploration beyond the reference pool.

## D.5 Efficiency Metrics

LLM inference dominates the computational cost of our framework, so efficiency primarily depends on model scale and hardware. For experiments with locally hosted LLMs, we report the average wall-clock time per successfully generated valid structure, including both generation and evaluation. We prioritize this metric over raw FLOPs because it directly captures the trade-off between higher per-step compute for larger models and their typically higher success rates. For experiments using API-accessed LLMs, we report token usage and estimate environmental impact in Appendix H. We compute the carbon footprint by converting total input and output tokens into energy consumption and equivalent $CO_2$ emissions using established energy-per-token factors.

| Measure | Definition | Parameter/Method | Reference Database |
|---|---|---|---|
| Metastability | Near-hull thermodynamic stability (proxy) | CHGNet-predicted $E_d < t$ eV/atom, $t \in \{0.0, 0.03, 0.1\}$ | MP (`2023-02-07-ppd-mp`) |
| Stability | DFT-verified on-hull stability | DFT-calculated $E_d \leq 0.0$ eV/atom (VASP) | MP (`2023-02-07-ppd-mp`) |
| Uniqueness | No duplicate within the generated set | `StructureMatcher` (default: `ltol`=0.2, `stol`=0.3 Å, `angle_tol`=5°) | Generated set |
| Compositional novelty | Reduced formula not in the reference database | Match by `reduced_formula` | MatBench-bandgap (106K) |
| Structural novelty | No structural match in the reference database | `StructureMatcher` (default: `ltol`=0.2, `stol`=0.3 Å, `angle_tol`=5°) | MatBench-bandgap (106K) |
| Novel polymorphs | Known composition, novel structure | Composition $\in$ MP; no structural match via `StructureMatcher` | Materials Project (MP) |
| S.U.N. rate (DiffCSP/FlowMM) | Fraction that is stable, unique, and novel | DFT stable + unique + novel (novelty against training set) | MP-20 (27K train, 45K total) |
| S.U.N. rate (MatLLMSearch) | Fraction that is stable, unique, and novel | DFT stable + unique + novel (novelty against reference database) | MatBench-bandgap (106K) |

Table S1: Evaluation metric definitions. Definitions of stability, metastability, matching criteria, and reference databases used for novelty.

## D.6 Summary

We summarize the key evaluation definitions used throughout this work in Table S1, including stability and metastability thresholds, reference databases for novelty evaluation, and the energy correction and compatibility settings applied.

# E Additional Experiments on CSD

We demonstrate the flexibility of our evolutionary pipeline by guiding LLMs to propose novel crystal structures with diverse mechanical characteristics. We evaluate five optimization strategies: (1) stability-oriented optimization ("Stability"), (2) property-oriented optimization ("Bulk Modulus"), (3) alternating multi-objective optimization ("Multi-turn"), (4) normalized weighted-sum optimization ("Weighted Sum"), and (5) lexicographic optimization ("Lexical"). As shown in Table S2, all strategies maintain high metastability rates, indicating that our framework can optimize target properties while preserving validity and stability.

| Model | Objective | Validity | | Metastability | | |
|---|---|---|---|---|---|---|
| | | | | M3GNet | CHGNet | |
| | | Structural | Composition | $E_d < 0.1$ | $E_d < 0.1$ | $E_d < 0.03$ |
| | Stability | 100.0% | 79.4% | 81.1% | 76.8% | 56.5% |
| | Bulk Modulus | 100.0% | 82.9% | 27.0% | 43.3% | 8.3% |
| MatLLMSearch (Llama 3.1-70B) | Multi-turn | 100.0% | 84.1% | 70.9% | 57.1% | 29.8% |
| | Weighted Sum | 100.0% | 88.1% | 74.0% | 59.8% | 36.5% |
| | Lexical | 100.0% | 89.5% | 84.7% | 78.0% | 59.9% |

Table S2: Compare experimental results under various optimization goals. We explore multi-objective optimization for stability and bulk modulus in three different ways.

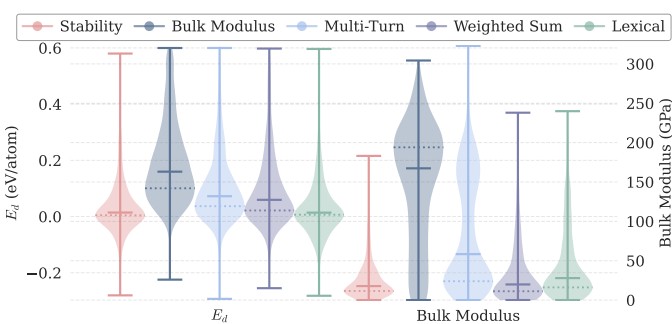

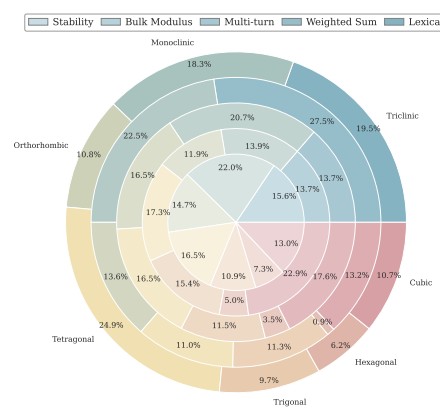

Figure S1: Comparison of optimization strategies targeting different objectives evaluated based on thermodynamic stability (decomposition energy $E_d$) and mechanical property (bulk modulus).

Figure S2: Crystal system distributions of generated structures under various optimization strategies. Structural diversity is preserved regardless of optimization objective.

| Bulk Modulus | Avg. Bulk Modulus (GPa) | | | | |
|---|---|---|---|---|---|
| Objective | Iter 1 | Iter 2 | Iter 3 | Iter 4 | Iter 5 |
| Target 100 GPa | $88.5 \pm 67.4$ | $91.5 \pm 42.7$ | $89.4 \pm 28.4$ | $98.7 \pm 25.5$ | $90.9 \pm 25.9$ |
| Maximize | $96.1 \pm 63.5$ | $123.7 \pm 40.9$ | $149.5 \pm 42.9$ | $171.8 \pm 50.8$ | $232.6 \pm 47.2$ |

Table S3: Bulk modulus controllability for CSD: (a) target 100 GPa and (b) maximize bulk modulus.

**Bulk modulus optimization.** To validate property-guided generation with MatLLMSearch, we perform single-objective optimization by changing the selection criterion from decomposition energy ($E_d$) to bulk modulus. In crystalline solids, bulk modulus is a key indicator of mechanical resistance to compression.

Bulk modulus values were computed using an Equation of State (EOS) workflow with the CHGNet potential. Structures are first relaxed with CHGNet to reach local energy minima. Isotropic volume perturbations were then applied to generate distorted structures. For each perturbation, a constrained ionic relaxation was performed to optimize atomic positions while fixing the cell dimensions, using a force convergence criterion of 0.1 eV/Å and a maximum of 500 optimization steps. Finally, the resulting energy-volume data were fitted to the Birch-Murnaghan EOS to extract the bulk modulus, which was then converted to GPa.

Figure S1 compares the distributions of decomposition energy ($E_d$) and bulk modulus across optimization strategies and highlights clear trade-offs. Bulk-modulus optimization shifts the generated structures toward higher bulk modulus values (peak density at 194 GPa versus 19 GPa for stability-oriented optimization), but this improvement comes with increased decomposition energy: the $E_d$ density peak shifts from 0.0 eV/atom (stability-oriented) to 0.1 eV/atom (bulk-modulus-oriented), indicating reduced thermodynamic stability.

**Bulk modulus controllability.** We evaluate CSD tasks that (a) target a specific bulk modulus and (b) maximize bulk modulus, as shown in Table S3. Experiments use DeepSeek-Reasoner with population size 10 for 5 iterations.

MatLLMSearch demonstrates effective controllability: in the 100 GPa targeting setting, the average bulk modulus approaches 98.7 GPa by iteration 4, while the maximum setting shows consistent improvement across iterations. These results highlight the flexibility of our framework for crystal structure design tasks with different objectives.

**Multi-objective optimization.** Beyond single-objective optimization, we explore multi-objective strategies that jointly consider thermodynamic stability and mechanical properties.

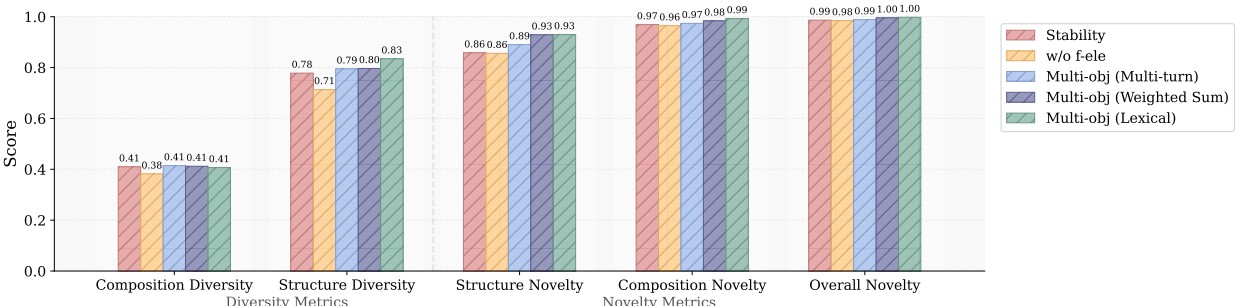

Figure S3: Diversity and novelty evaluation results for structures proposed under different experimental settings.

We first implement an alternating strategy ("Multi-turn"), which switches between stability and property optimization across iterations, with stability optimized first to provide a reliable starting point. The number of iterations allocated to each objective can be adjusted. As shown in Figure S1, this strategy achieves balanced performance, with $E_\text{d}$ centered around 0.037 eV/atom. The bulk modulus distribution suggests a stability–property trade-off, separating structures into higher-strength but moderately stable candidates versus highly stable but lower-strength candidates.

We also consider a normalized weighted-sum objective that combines both terms into a single scalar: $\mathcal{J} = w_e \cdot \hat{E}_\text{d} + w_b \cdot (1 - \hat{B})$, where $\hat{E}_\text{d}$ and $\hat{B}$ are min-max normalized values and $w_e = 0.7$, $w_b = 0.3$. This strategy produces structures with bulk modulus centered around 141 GPa and $E_\text{d}$ centered around 0.034 eV/atom.

Finally, the lexicographic ("Lexical") strategy treats stability as the primary criterion and considers bulk modulus only after a stability threshold is met (metastable structures with $E_\text{d} < 0.03$ eV/atom). This design penalizes low-stability candidates to keep stability dominant. While single-objective stability optimization achieves the highest metastability rate (76.8%), all multi-objective strategies maintain metastability rates above 50% while improving mechanical properties.

To ensure that optimization does not degrade generation quality, we also evaluate diversity and novelty under different objectives. Following prior work, we compute diversity and novelty on structures predicted to be metastable, using a structural distance cutoff of 0.1 and a composition distance cutoff of 2 for novelty. Results in Figure S3 show a consistent trade-off between property optimization and novelty: when explicitly targeting properties (e.g., bulk modulus), the model more often explores well-established stable chemical spaces, slightly reducing novelty while maintaining high diversity. Our evolutionary approach continues to encourage exploration of diverse structural motifs, as evidenced by the relatively uniform crystal system distributions in Figure S2.

## F   Additional Experiments on CSP

We further evaluate `MatLLMSearch` on crystal structure prediction across multiple compositions. Crystal structure prediction (CSP) aims to identify the lowest-energy atomic arrangement for a given composition.

For each CSP task, we first filter seed structures using compositional constraints and then run `MatLLMSearch` for 10 iterations. This setup enables the LLM to propose structures by leveraging optimized candidates with similar compositions.

To benchmark our framework, we compare against DiffCSP-generated candidates. For each composition, we use DiffCSP to sample 100 structures and then apply energy-guided optimization using an energy predictor trained on MP-20 for 1000 epochs, producing 10 optimized variants per initial sample. We evaluate energies after relaxation with Orb-v3 (Rhodes et al., 2025) for both methods.

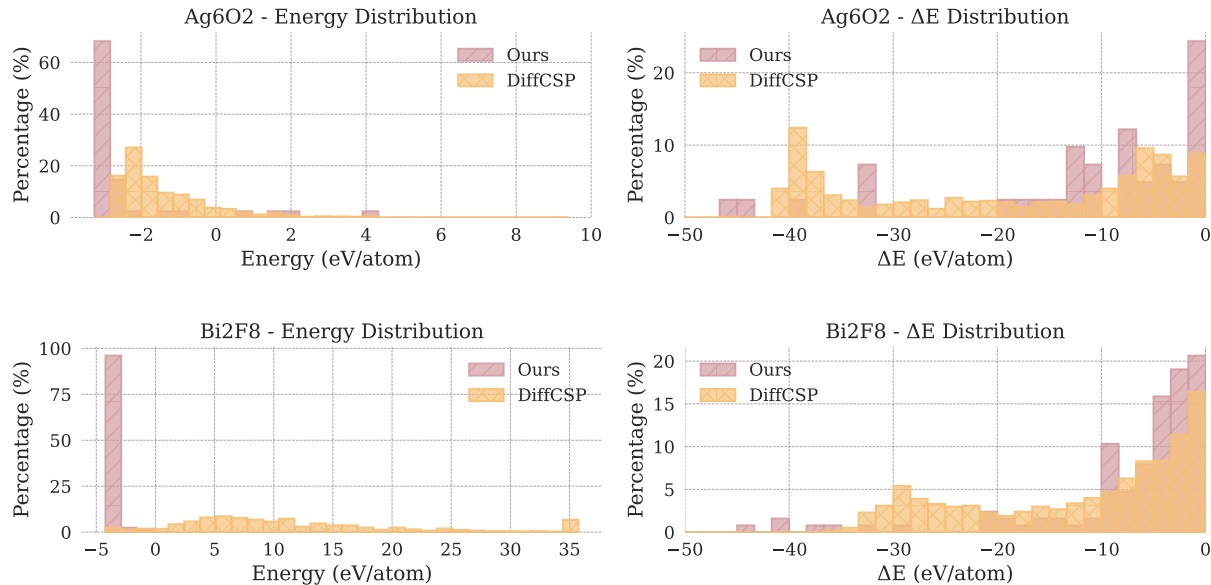

Figure S4: Energy and $\Delta E$ distributions comparing our LLM-generated structures versus DiffCSP predictions for $Ag_6O_2$ and $Bi_2F_8$ CSP tasks. LLM-generated structures show smaller $|\Delta E|$ values during relaxation than DiffCSP predictions.

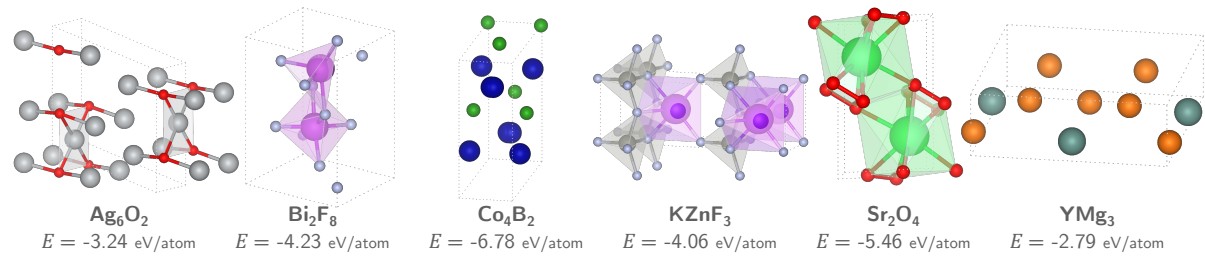

Figure S5: Representative CSP examples for six compositions ($Ag_6O_2$, $Bi_2F_8$, $Co_4B_2$, $KZnF_3$, $Sr_2O_4$, $YMg_3$) generated by MatLLMSearch.

MatLLMSearch successfully predicts structures for a range of compositions, including $Ag_6O_2$, $Bi_2F_8$, $Co_4B_2$, $KZnF_3$, $Sr_2O_4$, and $YMg_3$. Figure S5 shows representative predictions that achieve lower energies than the best DiffCSP candidates under the same evaluation protocol.

To further compare the two approaches, we analyze the energy and $\Delta E$ distributions. Figure S4 shows that many DiffCSP predictions undergo substantial changes during relaxation (large $|\Delta E|$), whereas our LLM-generated structures typically require smaller adjustments. This suggests that the initial configurations proposed by MatLLMSearch are closer to local energy minima.

## G    Hyperparameter Sensitivity Analysis for CSG

### G.1    Evolutionary Reproduction Hyperparameters

Our training-free evolutionary framework is relatively robust to hyperparameters compared to many traditional machine learning approaches. The reproduction phase introduces three key hyperparameters that control the LLM prompting context and sampling budget: population size ($K$), context size ($C$; number of parents), and children size ($c$; number of proposed offspring per prompt). Our baseline configuration ($C = 2$, $c = 5$) with

| Reproduction Configuration | # Unique / # Total Generated | $E_d < 0.1$ eV/atom | $E_d < 0.03$ eV/atom |
|:---:|:---:|:---:|:---:|
| $1 \rightarrow 5$ | 56.5% | 79.8% | 56.4% |
| $2 \rightarrow 5$ | 72.3% | 76.8% | 56.5% |
| $2 \rightarrow 2$ | 86.3% | 74.8% | 54.3% |
| $5 \rightarrow 5$ | 92.7% | 72.3% | 47.3% |
| $5 \rightarrow 2$ | 95.5% | 68.3% | 46.1% |

Table S4: Proportion of unique structures and their CHGNet-predicted metastability under varying reproduction configurations.

| LLM Temperature | # Unique / # Total Generated | $E_d < 0.1$ eV/atom | $E_d < 0.03$ eV/atom |
|:---:|:---:|:---:|:---:|
| 0.95 | 72.3% | 76.8% | 56.5% |
| 0.7 | 70.7% | 75.4% | 56.6% |
| 0.5 | 70.7% | 71.2% | 51.4% |
| 0.2 | 69.8% | 70.3% | 50.2% |

Table S5: Proportion of unique structures and their CHGNet-predicted metastability with different LLM temperatures.

Llama 3.1 (70B) achieves balanced performance, generating 72.29% unique structures while maintaining high stability.

Varying the parent-to-children ratio reveals a trade-off between diversity and stability. Increasing parent diversity ($C = 5$, $c = 2$) improves compositional uniqueness to 95.49% but slightly reduces stability (Table S4). In contrast, single-parent settings highlight the benefit of multi-parent crossover for maintaining structural diversity and stability. Overall, higher parent-to-children ratios can improve exploration quality, but the best setting depends on the desired balance between diversity and stability.

We also find that larger population sizes ($K$) can maintain stability and validity comparable to smaller populations. Increasing $K$ increases diversity within each iteration, which can reduce the overrepresentation of $f$-electron elements and broaden compositional coverage. However, larger populations can also increase duplication across iterations, suggesting that earlier stopping or stronger de-duplication may be beneficial. These observations enable application-specific tuning of the reproduction hyperparameters.

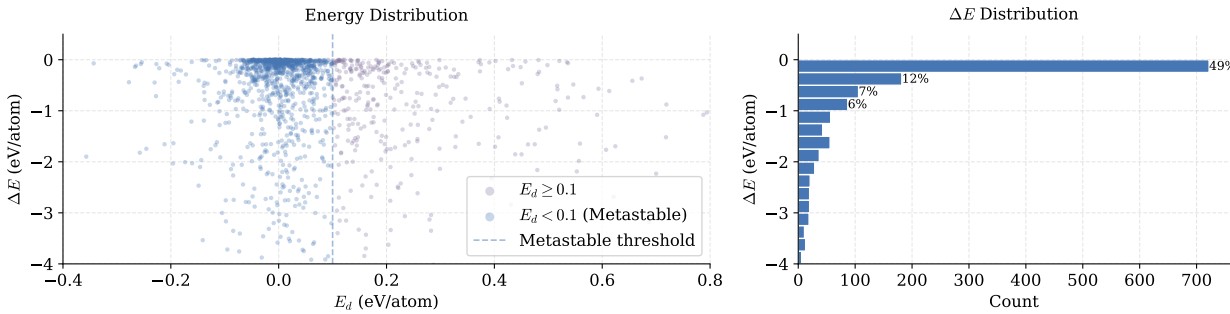

Figure S6: Distribution of energy change $\Delta E$ before/after structural relaxation and decomposition energy ($E_d$) for structures proposed by LLM, evaluated using the pretrained CHGNet.

| Method | Primary Format | Generative | Model | Training |
|---|---|---|---|---|
| CDVAE (Xie et al., 2022) | 3D | Diffusion | GNN | Training |
| MatterGen (Zeni et al., 2025) | 3D | Diffusion | GNN | Training |
| Flam-Shepherd and Aspuru-Guzik (2023) | 3D | AR | Transformer | Training |
| DiffCSP (Jiao et al., 2024) | 3D | Diffusion | GNN | Training |
| CrystalTextLLM (Gruver et al., 2024) | Text/CIF | LLM | Transformer | Fine-tuning |
| FlowMM (Sriram et al., 2024) | 3D | Flow | GNN | Training |
| MatLLMSearch (Ours) | Text/CIF/POSCAR | LLM | Llama 3.1 | N/A |

Table S6: A collection of generative models on computational materials discovery. Training denotes if training/fine-tuning is required on materials databases.

| Format | # Unique / # Total Generated | $E_d < 0.1$ eV/atom | $E_d < 0.03$ eV/atom |
|---|---|---|---|
| POSCAR (4) | 76.7% | 75.4% | 55.3% |
| POSCAR (12) | 72.3% | 76.8% | 56.5% |
| CIF | 75.1% | 68.9% | 49.5% |

Table S7: Proportion of unique structures and their CHGNet-predicted metastability using different structure formats.

## G.2 Effect of Structure Relaxation

To quantify the role of structural relaxation in our framework, we define $\Delta E$ as the CHGNet energy difference after versus before relaxation. Figure S6 shows that most LLM-proposed structures exhibit relatively small changes: 61.2% have $|\Delta E| < 0.5$ eV/atom. This indicates that the generated structures are often already close to local energy minima and typically require only modest refinement during relaxation.

## G.3 Effect of Structure Representation

As summarized in Table S6, most machine-learning approaches to crystal structure generation operate on 3D representations (e.g., graphs or periodic coordinate sets) using GNNs or Transformers with diffusion or autoregressive generators. In contrast, LLM-based approaches require a text serialization of the crystal structure.

We therefore study how the structure serialization affects generation efficiency and quality. Specifically, we compare CIF and POSCAR formats, with POSCAR coordinates written at either 4 or 12 decimal places. Examples are shown in Figure S7.

We first analyze token efficiency by measuring token length distributions on MatBench (Figure S8). POSCAR with 4 decimal places is the most token-efficient, followed by POSCAR with 12 decimal places; CIF is the least token-efficient due to its more verbose formatting and metadata.

Table S7 shows that POSCAR with 12 decimal places achieves slightly higher (meta)stability under both thresholds ($E_d < 0.03$ and 0.1 eV/atom). We therefore use POSCAR with 12 decimal places as a practical trade-off between token efficiency and numerical fidelity. Differences across formats are modest overall, suggesting that our approach is relatively robust to the choice of serialization and that pre-training exposure to crystallographic formats may reduce sensitivity.

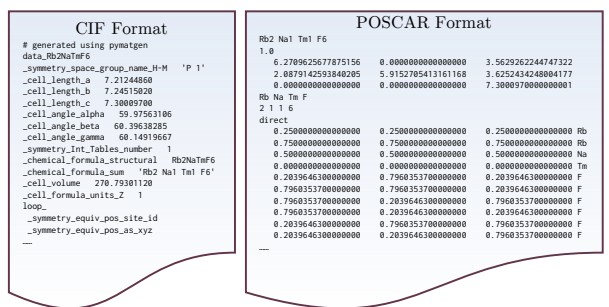

Figure S7: Structure string examples of CIF format and POSCAR format.

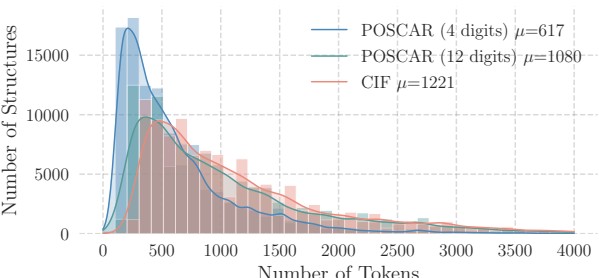

Figure S8: Token efficiency comparison under CIF formatting and POSCAR formatting for the precision of 4 and 12 decimal. $\mu$ indicate the mean of token lengths.

| Base Model | Fine-Tuning? | Prompting Strategy | Metastability (%) (CHGNet) | |
|---|---|---|---|---|
| | | | $E_{\mathrm{d}} < 0.1$ eV/atom | $E_{\mathrm{d}} < 0.03$ eV/atom |
| Llama 3.1 8B | ✗ | Zero-shot | 0.0 | 0.0 |
| | | MatLLMSearch | 27.7 | 10.0 |
| | ✓(8 bit) | Zero-shot | 0.0 | 0.0 |
| | | MatLLMSearch | 45.5 | 22.7 |
| Llama 3.1 70B | ✗ | Zero-shot | 25.8 | 12.9 |
| | | MatLLMSearch | 76.8 | 56.5 |
| | ✓(4 bit) | Zero-shot | 13.9 | 2.8 |
| | | MatLLMSearch | 66.0 | 48.0 |

Table S8: Meta-stability comparison of prompting strategy across models with and without fine-tuning.

## G.4 Effect of Base LLM Choice

### G.4.1 Within-Family Comparison (Llama 3.1)

**Model scale.** To study the effect of base LLM size, we evaluate MatLLMSearch with Llama 3.1 models at 8B and 70B parameters, including both pre-trained and fine-tuned variants. Table S8 shows that model capability strongly affects CSG metastability: the 70B model achieves 76.8% metastability in our full framework compared to 27.7% for the 8B model, suggesting that useful crystallographic priors emerge only at sufficient scale.

**Fine-tuning integration.** Fine-tuned models show substantial improvements when integrated with our evolutionary framework. The 8B fine-tuned model achieves 45.5% metastability (from 27.7% baseline), while the 4-bit quantized 70B model maintains 66.0% metastability despite compression constraints. See Table S8 for detailed results and prompting strategies. Importantly, our information value metric demonstrates that both fine-tuned and pre-trained models integrate seamlessly into the evolutionary framework, with performance scaling according to base model capability.

**Generation strategy.** We compare two generation strategies: zero-shot prompting (without reference structures or iterative evolution) and our evolutionary framework. Zero-shot prompting fails for the 8B model and reaches only 25.8% metastability for the 70B model. Our evolutionary framework substantially improves performance by systematically exploring chemical space while maintaining physical validity. Reference structures accelerate convergence and promote diverse exploration, while iterative evolution is essential for both the quantity and quality of valid generations.

**Model temperature.** The temperature hyper-parameter controls sampling randomness in language models by scaling the logits before softmax transformation. Higher temperatures flatten the probability distribution, increasing sampling diversity, while lower temperatures concentrate probability mass on the most likely tokens. While temperature is commonly associated with model creativity, with higher temperatures generally

| Method | Validity | Metastability ($E_d < 0.1$, **CHGNet**) | Metastability ($E_d < 0.0$, **CHGNet**) | S.U.N. **(CHGNet)** |
|---|---|---|---|---|
| GPT-5-mini | 98.45% | 74.60% | 50.05% | 46.24% |
| GPT-5-chat | 98.12% | 64.36% | 46.93% | 44.37% |
| GPT-5 | 99.63% | 88.33% | **63.22%** | **55.31%** |
| Grok-4 | **99.92%** | 87.13% | 60.29% | 49.80% |
| DeepSeek Reasoner | 99.25% | **88.90%** | 61.22% | 48.25% |
| Claude Sonnet 4.5 | 99.10% | 78.71% | 50.21% | 38.99% |

Table S9: Base LLM comparison using MatLLMSearch for CSG (population $K = 100$, 10 iterations). Unless noted otherwise, all models use temperature 1.0 and max tokens 8000; metrics are computed with CHGNet.

producing slightly more novel outputs (Peeperkorn et al., 2024), this relationship remains an active area of research.

Crystal structure generation requires exploring diverse candidates while maintaining physical validity. We use temperature 0.95 in our baseline to encourage exploration while preserving stability. Table S5 reports CHGNet-evaluated metastability across temperatures. At 0.95, the model generates 76.81% metastable structures under $E_d < 0.1$ eV/atom; lowering temperature to 0.7 yields 75.38%, and 0.5 yields 71.18%. Under the stricter threshold $E_d < 0.03$ eV/atom, the corresponding rates at 0.95, 0.7, 0.5, and 0.2 are 56.5%, 56.6%, 51.4%, and 50.2%, respectively. Overall, stability remains high across settings, indicating that the pipeline is robust to temperature variation.

### G.4.2 Across-Family Comparison

We extend the base LLM evaluation to six models across multiple families (DeepSeek, GPT, Grok, Claude) in Table S9. We compare those LLMs under the same evolutionary setup and report: (1) validity, the fraction of generations that parse into physically valid periodic structures; (2) metastability under $E_d < 0.1$ eV/atom (CHGNet); (3) stability under $E_d < 0.0$ eV/atom (CHGNet); and (4) S.U.N. rate, the fraction of generations that are simultaneously stable ($E_d < 0.0$ eV/atom by CHGNet), unique within the generated set (StructureMatcher), and novel with respect to the MatBench-bandgap reference set (no match via StructureMatcher; we additionally require reduced-formula novelty for a more conservative estimate).

Table S9 shows consistently high validity across models, while metastability and S.U.N. vary more substantially. GPT-5 and DeepSeek Reasoner achieve the strongest overall performance, indicating that the framework benefits from both robust instruction-following (validity) and domain-specific priors (stability and novelty).

### G.5 Scaling with Population Size

We increase the population size to $K = 500$ to study the scalability of LLM-guided evolution. We report metastability and S.U.N. (computed based on metastability, with novelty assessed against MatBench-bandgap) in Table S10. When increasing $K$ from 100 to 500, the number of valid structures scales by roughly 5×, demonstrating that MatLLMSearch can scale to high-throughput settings. For DeepSeek Reasoner, metastability and S.U.N. decrease slightly at larger $K$. However, metastability remains substantially higher than diffusion baselines, indicating that LLM-guided evolution preserves thermodynamic validity even at scale. We visualize compositional overlap with the reference database in Figure 5. The scattered but mostly overlapping distribution suggests an exploitation tendency toward known compositions.

### G.6 Runtime and Resource Cost

Efficiency depends primarily on the base model and hardware (see Appendix D for metric definitions). For local LLMs, we report average wall-clock time per valid unique structure, including both generation and evaluation. With Llama-3.1-70B-Instruct on 4×A6000 GPUs and population size $K = 100$, this cost is 62.35 s per valid unique structure. We note that although CrystalTextLLM reports 51.6 s per valid structure under

| Population | Validity | Metastability ($E_\mathbf{d} < 0.1$, **CHGNet**) | Metastability ($E_\mathbf{d} < 0.0$, **CHGNet**) | S.U.N. (**CHGNet**) |
|---|---|---|---|---|
| $K = 100$ | **99.25%** | **88.90%** | **61.22%** | **48.25%** |
| $K = 500$ | 98.97% | 84.31% | 55.51% | 43.59% |

Table S10: Population scaling using MatLLMSearch for CSG (10 iterations; DeepSeek Reasoner as the base LLM across rows).

| Model | Energy (kWh / 1,000 tokens) | Total Tokens (M) | Energy (kWh) | $CO_2$ (kg) |
|---|---|---|---|---|
| Claude Sonnet 4.5 | 0.00139 | 4.72 | 6.56 | 2.92 |
| DeepSeek Reasoner | 0.01450 | 8.04 | 116.52 | 51.85 |
| GPT-5 | 0.00326 | 7.06 | 22.98 | 10.22 |
| GPT-5-chat | 0.00326 | 2.73 | 8.90 | 3.96 |
| GPT-5-mini | 0.00071 | 5.79 | 4.10 | 1.83 |
| Grok-4 | 0.00605 | 7.25 | 43.89 | 19.53 |

Table S12: Estimated carbon footprint for LLM API calling.

its reported setting, wall-clock numbers should be interpreted cautiously because hardware, decoding settings, and evaluation pipelines can differ.

**Generation vs. evaluation.** Table S11 breaks the cost into generation and evaluation components for 8B versus 70B local models.

**Inference cost rationale.** We prioritize time per successful sample (rather than FLOPs) because larger models often have higher validity, which can reduce the amortized cost per valid structure despite higher per-step compute.

**Carbon footprint.** In Table S12, we estimate energy use and $CO_2$ emissions for API-based LLM calls by converting total input and output tokens using inferred energy-per-token factors (Jegham et al., 2025).

| Model | Avg. Generation Time (s) | Avg. Evaluation Time (s) |
|---|---|---|
| 70B | 55.99 | 6.36 |
| 8B | 53.40 | 10.80 |

Table S11: Generation/evaluation time breakdown per valid unique structure for local Llama 3.1 models ($K = 100$).

# H Additional Analysis on CSG

## H.1 CSG Trajectory

We visualize a representative stability-oriented optimization trajectory in Figure S9 (Llama-3.1-70B), illustrating how candidate structures are iteratively refined into more (meta)stable variants.

## H.2 Extended Novelty Verification

To address concerns about data leakage and memorization, we perform an additional novelty check against the Materials Project (MP; >200,000 structures) using compositional and structural matching.

**Novelty criteria.** We verify novelty against MP using two criteria: *compositional novelty* and *structural novelty* (StructureMatcher default settings: ltol=0.2, stol=0.3 Å, angle_tol=5°). We label a structure as *overall novel* only if both its reduced composition is absent from MP and it has no structural match in MP, providing a conservative estimate. We also report *novel polymorphs*, i.e., structures whose compositions exist in MP but whose crystal structures do not match any MP entry.

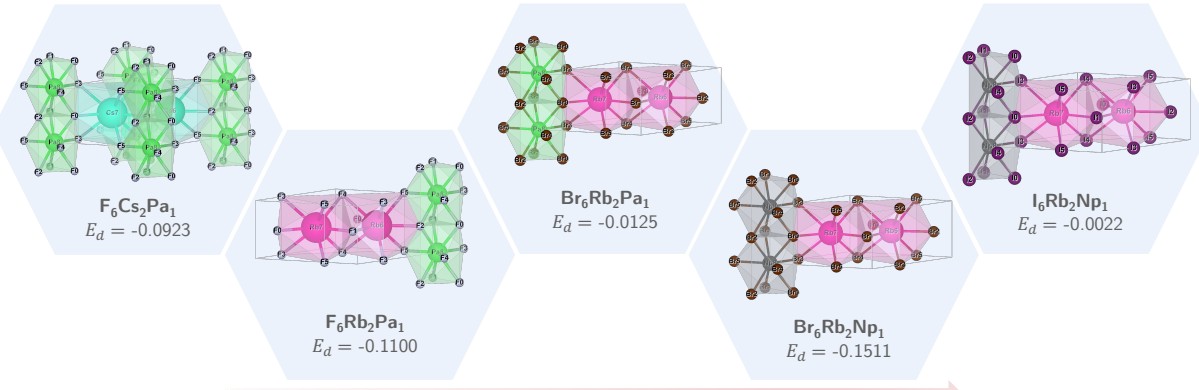

Figure S9: Example evolutionary trajectories of crystal structure generation.

| Configuration | Overall Novelty | Composition in MP | Structure Matched |
|---|---|---|---|
| MatLLMSearch (Ref + Iter) | 69.2% | 30.8% | 18.5% |
| No Ref + Iter | **91.4%** | **8.6%** | **1.8%** |
| Ref + No Iter | 72.6% | 27.4% | 16.7% |
| No Ref + No Iter | 56.7% | 43.3% | 5.5% |

Table S13: Novelty evaluation against MP for ablations (Llama-3.1-70B, $K = 100$). "Ref" indicates using reference structures; "Iter" indicates iterative evolution.

**Findings.** We acknowledge that completely ruling out data leakage is impossible for LLMs trained on broad scientific corpora. Nevertheless, Table S13 shows that the "No Ref + Iter" ablation attains the highest overall novelty and the lowest MP overlap, suggesting that high novelty is not simply driven by copying reference-set structures.

We further run the same MP-based novelty verification for a larger-scale experiment with a different base LLM. Across settings, fewer than 30% of generated structures share compositions with MP; among those composition-overlapping cases, most have no structural match (RMSD $> 0.3$ Å under `StructureMatcher`), yielding 8–12% novel polymorphs.

In Table S14, we summarize MP-based novelty statistics at two scales. In both cases, most generations are compositionally novel relative to MP, and more than 95% of generated compositions are unique within the generated set. We also observe a non-trivial fraction of novel polymorphs (composition overlaps with MP but no structural match), indicating that the framework can propose new arrangements even when compositions are known. The larger-scale run shows slightly higher compositional and structural novelty rates, consistent with broader sampling at increased population size.

| Categories | Llama-3.1-70B ($K = 100$) | DeepSeek-Reasoner ($K = 100$) | DeepSeek-Reasoner ($K = 500$) | Definition |
|---|---|---|---|---|
| Total structures | 1,479 | 1,604 | 8,602 | Total generated structures |
| Compositional diversity | 1,417 | 1,572 | 8,444 | Unique compositions among total generated |
| Compositional novelty | 1,023 | 1,188 | 6,568 | Composition $\notin$ MP |
| Structural novelty | 1,205 | 1,337 | 7,260 | No structure match in MP |
| Compositions in MP | 456 | 416 | 2,034 | Composition $\in$ MP |
| Structures matched | 274 | 267 | 1,342 | Composition $\in$ MP, structure match MP entries |
| Novel polymorphs | 182 | 149 | 692 | Composition $\in$ MP, no structure match in MP |

Table S14: Novelty analysis: comparison between the original experiment (Llama-3.1-70B, $K = 100$), DeepSeek-Reasoner ($K = 100$), and larger-scale experiment (DeepSeek-Reasoner, $K = 500$).

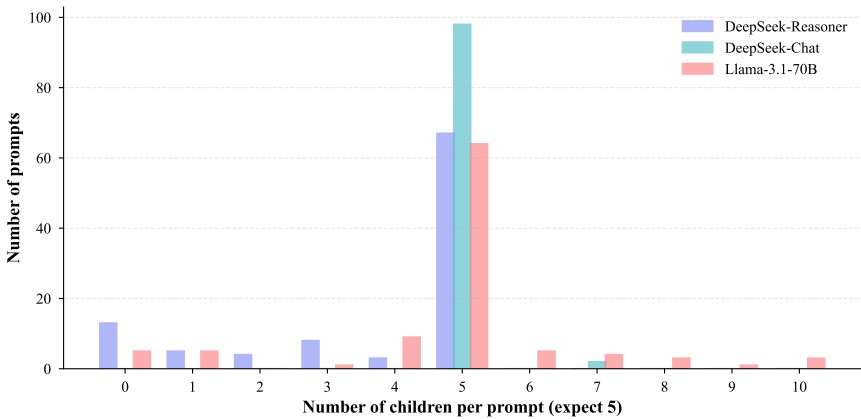

Figure S10: Distribution of children generated per prompt (before parsing and validation) for three LLM configurations.

| Category | DeepSeek-Reasoner | DeepSeek-Chat | Llama-3.1-70B |
|---|---|---|---|
| *Generation Completeness (out of 100 prompts)* | | | |
| Prompts with $\neq 5$ children | 33 | 2 | 36 |
| *JSON Format Error (out of 100 prompts)* | | | |
| Empty responses | 12 | 0 | 2 |
| JSON parsing error | 21 | 0 | 56 |
| **Total Regex Matched Structures** | **405** | **504** | **496** |
| POSCAR parsing failures (after automatic fixing) | 9 | 0 | 40 |
| **Total Parsed Structures** | **396** | **504** | **456** |
| *POSCAR Format Error (before automatic fixing, non-exclusive)* | | | |
| Truncated strings (atom count mismatch) | 15 | 19 | 237 |
| Invalid numeric values | 0 | 0 | 26 |
| Invalid element tokens | 0 | 0 | 20 |
| *Validation Failures (out of total parsed)* | | | |
| Overlapping atoms ($<0.5\,\text{Å}$) | 0 | 13 | 125 |
| Invalid stoichiometry | 0 | 0 | 0 |
| Charge imbalance | 0 | 0 | 0 |
| Volume/periodicity errors | 0 | 0 | 0 |
| Relaxation errors | 0 | 0 | 3 |
| $E_\text{d}$ calculation errors | 0 | 1 | 2 |
| **Valid Structures** | **349** | **428** | **308** |

Table S15: Failure mode breakdown (100 prompts $\times$ 5 children).

## I  Failure Mode Analysis

We conduct a failure analysis of LLM-guided crystal structure generation to identify where and why it fails. Using 100 prompts (2 parent structures each, 5 requested children per prompt) for three LLM configurations (DeepSeek-Reasoner with 16,000 tokens for extended reasoning, DeepSeek-Chat with standard 8,000 tokens chat, and Llama-3.1-70B baseline), we track all 500 expected structures per model over multiple validation stages: LLM generation, parsing, basic validation, CHGNet relaxation, and $E_\text{d}$ calculation.

Figure S10 shows the distribution of the number of children generated per prompt with regex matching applied. DeepSeek-Chat demonstrates good instruction following for the 5-child requirement, while Llama-3.1-70B

shows difficulty producing complete responses. DeepSeek-Reasoner requires extra tokens for reasoning, it may produce empty responses.

**Format error.** Format errors in Appendix I include JSON parsing errors when handling each response and POSCAR parsing errors when handling the POSCAR string for each child inside the JSON response.

*JSON Format Errors* include empty responses and malformed JSON. DeepSeek-Chat shows the best overall instruction-following ability for formatting. DeepSeek-Reasoner can use up tokens in reasoning, leading to empty responses. Llama more often generates malformed JSON. We apply regex matching to recover structures from malformed JSON, successfully extracting 405, 504, and 496 structures, respectively.

*POSCAR Format Errors* are counted non-exclusively: truncated strings (atom count mismatches before fixing), invalid numeric values (e.g., "0.qlBay44"), and invalid element tokens (e.g., "Ue"). Truncation can occur mid-response (not only at the end), indicating LLM truncation behavior beyond token limits. Minor count mismatches are auto-corrected by aligning declared counts with coordinate lines. E.g. given a structure declaring "Lu 6 B 18 Rh 18" but providing only 14 B and 14 Rh coordinates, we adjust the atom counts to align with the actual coordinate lines.

**Validation failures.** DeepSeek models show better performance in generating valid structures than Llama-3.1-70B, which more often generates structures with overlapping atoms or severe structural errors that lead to relaxation or $E_d$ calculation errors.

## J  Mutation and Crossover Analysis

To characterize genetic operations performed by LLMs during two-parent structure generation, we analyze approximately 1K parsed valid structures from 100 prompts with two parent structures each. We classify operations into *mutation* (single-parent inheritance) and *crossover* (multi-parent recombination).

### J.1  Mutation Analysis

1. **Same-group substitution:** Measures whether child elements belong to the same periodic group as replaced parent elements (e.g., $Dy_5RuI_7 \rightarrow Dy_5RuBr_7$ where $I \rightarrow Br$ within Group 17 halogens).

2. **Stoichiometry preservation:** Evaluates exact reduced formula matching using pymatgen composition equality (e.g., $YbAlB_{14} \rightarrow ZrAlB_{14}$ preserves a 1:1:14 ratio).

3. **Composition similarity:** Quantifies element overlap via Jaccard index (set intersection/union) and Magpie feature-based cosine similarity (e.g., $Dy_2Sn_5Rh_3$ vs $Er_2Sn_5Rh_3$ yields Jaccard = 0.50, similarity = 0.993 due to shared Sn and Rh).

4. **Space group preservation:** Checks if the child maintains the parent's crystallographic space group number.

5. **Symmetry preservation:** Evaluates preservation of both space group and Wyckoff positions.

6. **Lattice similarity:** Assesses volume ratios and lattice parameter deviations (higher volume ratio indicates more similar cell volumes).

7. **Structural matching:** Full structural matching using `StructureMatcher` with `FrameworkComparator` to detect exact geometric isomorphism regardless of element identity.

**Observations.** DeepSeek models in general demonstrate systematic application of crystallographic knowledge, supported by the high substitution rate of same-group elements as well as space group and symmetry preservation. All models exhibit minimal stoichiometry preservation and 0 structural matches, preferring compositional and structural exploration over copying or minor modifications. Both DeepSeek models show a slight preference for Parent 0 (the first parent in the prompt).

| Category | DeepSeek-Reasoner | | DeepSeek-Chat | | Llama-3.1-70B | |
|---|---|---|---|---|---|---|
| | **P0** | **P1** | **P0** | **P1** | **P0** | **P1** |
| *Mutation Analysis (Per-Parent Inheritance)* | | | | | | |
| 1. Same-group elemental substitution (%) | 63.3 | 62.3 | 47.3 | 52.5 | 43.9 | 51.1 |
| 2. Stoichiometry preservation (%) | 0.0 | 0.0 | 1.3 | 0.6 | 2.2 | 1.7 |
| 3. Composition similarity (normalized) | 0.821 | 0.787 | 0.826 | 0.802 | 0.774 | 0.770 |
| 4. Space group preservation (%) | 53.1 | 41.8 | 39.8 | 32.3 | 18.9 | 27.8 |
| 5. Symmetry preservation (%) | 52.6 | 40.8 | 37.0 | 28.6 | 15.6 | 16.7 |
| 6. Lattice similarity (volume ratio) | 0.783 | 0.681 | 0.760 | 0.676 | 0.597 | 0.654 |
| 7. Structural matching (%) | 0.0 | 0.0 | 0.0 | 0.0 | 0.0 | 0.0 |
| *Crossover Analysis (Multi-Parent Operations)* | | | | | | |
| 8. Group-based recombination (%) | 18.2 | | **31.0** | | 14.4 | |
| 9. Ratio bounds (%) | 100.0 | | 100.0 | | 100.0 | |
| 10. Density bounds (%) | 79.0 | | 79.1 | | 52.2 | |
| *Parent Influence Analysis* | | | | | | |
| 11. Dominant parent distribution (P0/P1) | 58.7 / 41.3 | | 58.9 / 41.1 | | 47.8 / 52.2 | |

Table S16: Mutation and crossover analysis results. Mutation analysis shows per-parent inheritance metrics (Parent 0 and Parent 1). Crossover analysis shows global multi-parent crossover metrics.

## J.2 Crossover Analysis

8. **Group-based recombination:** Identifies children incorporating unique elements from **both** parents (e.g., $AgRhF_6 + ZrAlAu_2 \rightarrow AlRhAu_2$ where the child takes a Ag→Au substitution from Parent 0 and Al, Au elements from Parent 1, achieving true multi-parent crossover).

9. **Ratio bounds:** Verifies whether child element fractions lie between parent fractions for shared elements (e.g., in $AlRhAu_2$, the Au fraction (0.50) lies within Parent 0's 0.00 and Parent 1's 0.50 range, demonstrating compositional interpolation).

10. **Density bounds:** Assesses if child density falls within $\pm 20\%$ of the parent density range.

11. **Dominant parent distribution:** Calculated by comparing each child structure to both parent structures by averaging per-parent similarity scores from mutation analysis. The parent with higher similarity is determined as the dominant parent. The distribution reports the proportion of children dominated by each parent.

**Observations.** DeepSeek-Chat achieves 31.0% group-based recombination rate, exceeding the other two models, suggesting better multi-parent exploration. All models follow a universal compositional interpolation constraint suggested by the perfect ratio bounding. DeepSeek models maintain 79% physically reasonable densities, while Llama shows higher average deviation indicating more often generation of geometrically unrealistic structures, which is accompanied by its higher composition error rate (Appendix I).

