# OpenReview forum: "MatLLMSearch: Crystal Structure Discovery with Evolution-Guided Large Language Models"
_TMLR — Rejected by TMLR_

### Review · Reviewer_dFDA · 2025-11-12

**Summary Of Contributions:**

This article addresses the challenge of crystal structure generation (CSG) in materials science, which is crucial for discovering new materials with specific properties. Prior to this work, existing methods often relied on extensive fine-tuning of large language models (LLMs) on materials databases, limiting their efficiency and accessibility. The authors propose MatLLMSearch, a framework that utilizes pre-trained LLMs as intelligent agents within an evolutionary algorithm to generate stable crystal structures without additional fine-tuning. They found that their approach achieved a 78.38% metastable rate and a 31.7% stability rate verified by density functional theory (DFT), outperforming specialized models like CrystalTextLLM. This work reveals that pre-trained LLMs can effectively generate valid and stable crystal structures, providing a more efficient and accessible method for materials discovery.

**Strengths**:

- The proposed approach of prompting large language models (LLMs) to generate new materials without fine-tuning is interesting and potentially impactful.

- The paper is well-written, clear, and easy to follow.

**Weaknesses**:

- **Novelty of generated structures**: It is unclear how the authors verify that the generated structures are truly novel rather than comparing them with the $K \times P$ structures from the initial generation pool. A more rigorous novelty check (e.g., structural or compositional similarity metrics against known databases) would strengthen the claims to make sure that the model is not generating structures or compositions potentially memorized from pre-training data or accessed from the internet.

- **Fairness of comparison**: The results comparison appears unfair. The paper reports outperforming a fine-tuned Llama-70B model (CrystalTextLLM), yet the evaluation settings differ significantly (10K samples vs. 1,479). It is difficult to attribute performance differences solely to the proposed method under such conditions.

- **Missing baseline**: The paper does not include results using CrystalTextLLM-70B (Gruver et al., 2024) as a base model within the proposed prompting framework. Since the novelty primarily lies in the proposed generation process rather than the choice of base model, such a comparison is essential to support the paper’s claims. Hence, it would be unfair/unreasonable to claim that LLM inference only-approaches outperform domain-specific fine-tuned LLMs approaches based on the reported results.

- **Redundant claims**: CrystalTextLLM has already demonstrated that LLMs can effectively propose small, meaningful modifications to existing materials—precisely the mechanism underlying this paper’s mutation and crossover strategy. This overlap raises concerns about the degree of conceptual novelty.

- **Lack of failure analysis**: The paper would benefit from a discussion of failure cases or hallucinations, which are common in this domain (as shown in works such as CrystalTextLLM and LLM4Mat-Bench (Rubungo et al., 2024)). Acknowledging these limitations would make the contribution more balanced and credible.

- **Overall novelty**: While the idea of prompting LLMs for material generation is appealing, the methodological and empirical novelty seems insufficient for TMLR, given the overlap with existing approaches and limited depth of new insights.

**Audience:**

Yes

**Audience Explanation:**

LLMs for materials discovery represent an emerging research direction that is gaining significant attention in both the machine learning and materials science communities.

**Claims And Evidence:**

No

**Claims Explanation:**

- The paper lacks rigorous verification of the novelty of generated structures and does not rule out potential memorization from pretraining data.

- The experimental comparison is not fair or comprehensive—key baselines like CrystalTextLLM-70B are missing, and evaluation settings differ significantly.

- The work shows limited conceptual novelty and lacks discussion of failure cases or hallucinations observed in related studies.

**Requested Changes:**

- The authors should incorporate a rigorous novelty verification process—such as structural or compositional similarity checks against established materials databases—to ensure the generated structures are genuinely new and not memorized from pretraining data.

- The authors should standardize the evaluation settings (e.g., sample sizes and datasets) to enable a fair comparison with prior models like CrystalTextLLM and ensure that reported performance gains are attributable to the proposed method.

- The authors should include CrystalTextLLM-70B as a baseline within their proposed prompting framework to more accurately assess the contribution of their generation process independent of the base model.

- The authors should better differentiate their method from prior work, clearly articulating what conceptual or methodological advances it provides beyond the mutation and crossover strategies already explored in CrystalTextLLM.

- The authors should analyze and discuss failure cases or hallucinations in their generation process to provide a balanced assessment and highlight the method’s current limitations.

---

> ### Author Response · Authors · 2025-12-06
> **Response to Reviewer dFDA (1/2)**
>
> Thank you for your feedback. We take all concerns seriously and have conducted additional experiments and analyses to address  each.
>
> ### **1. Novelty**
>
> We respectfully but strongly disagree with the assessment that "the methodological and empirical novelty seems insufficient for TMLR." This assessment misaligns with TMLR's explicit acceptance criteria. According to the official TMLR acceptance criteria (https://jmlr.org/tmlr/acceptance-criteria.html):
>
> > "Crucially, it should not be used as a reason to reject work that isn't considered 'significant' or 'impactful' because it isn't achieving a new state-of-the-art on some benchmark. **Nor should it form the basis for rejecting work on a method considered not 'novel enough', as novelty of the studied method is not a necessary criteria for acceptance.**"
>
> Since Reviewer dFDA acknowledged that our work is of interest to TMLR's audience, and TMLR explicitly states that novelty is not a necessary criterion, the novelty concern should not be grounds for rejection.
>
> **New Perspective and Methodological Contributions**
>
> Our work introduces a fundamentally different perspective: pre-trained LLMs as intelligent agents rather than tools requiring fine-tuning. While CrystalTextLLM fine-tunes LLMs on materials databases, we demonstrate that pre-trained LLMs already possess rich embedded scientific knowledge from vast scientific corpora. This represents a paradigmatic shift in how we conceptualize LLMs for materials discovery.
>
> Our claims are well supported by experiments and strengthened by new experiments and analysis conducted in response to all reviewers' feedback:
>
> 1. **Multi-LLM Validation**: We evaluated MatLLMSearch with LLMs across multiple model families (Llama-3.1-70B, GPT-5, Grok-4, DeepSeek-Reasoner, Claude Sonnet 4.5, GPT-5-mini, GPT-5-chat). High validity and metastability rate of generated structures are observed in multiple LLMs, demonstrating robust generalization.
>
> 2. **Larger-Scale Experiment**: We conducted a scaling study with population $K=500$ (DeepSeek-Reasoner), generating 8,602 structures, which is significantly larger than prior evaluations. It demonstrates linear scaling in quantity with minor quality drop, showing the framework scales effectively from small to large generation scales.
>
> 3. **S.U.N. Evaluation**: For novelty assessment, we performed a clear S.U.N. calculation comparison and applied a more strict S.U.N. evaluation that compares against the entire MatBench-bandgap dataset (~106K structures).
>
> 4. **Comprehensive Data Leakage Analysis**: We conducted rigorous novelty verification against Materials Project (over 200,000 structures) using compositional matching (exact formula) and structural matching (RMSD < 0.3Å). Our main results with Llama-3.1-70B demonstrate 69.2% novelty composition no overlap in MP. providing strong quantitative evidence against systematic memorization.
>
> 5. **Comprehensive Ablation Studies**: We performed extensive ablation experiments verifying framework components (with/without references, with/without iteration), including novelty verification for each configuration, analyzing the contribution of each component.
>
> 6. **Multi-Task Flexibility**: We demonstrate MatLLMSearch effectiveness across CSG, CSP, and multi-objective optimization without task-specific fine-tuning, including discovery of novel metastable $Na_3AlCl_6$ polymorphs.
>
> 7. **Extensive Limitations Discussion**: We provide thorough analysis of framework limitations, failure modes, and scalability considerations throughout the manuscript.

---

> ### Author Response · Authors · 2025-12-06
> **Response to Reviewer dFDA (2/2)**
>
> ### **2. Comparison with CrystalTextLLM**
>
> **Fairness of Comparison**: We provided comprehensive evaluation over Diversity, Stability, Novelty in the manuscript, clearly attributing performance differences to methodology vs. sample size effects. We also extended the comparison with standardized evaluation of larger-scale experiment in Appendix F.2
>
> **CrystalTextLLM-70B Baseline**: We have presented the suggested experiments on CrystalTextLLM in Appendix F.1 of the draft. We integrated CrystalTextLLM models (8B and 70B) as base models within our MatLLMSearch framework, to isolate the contribution of the evolutionary generation process independent of base model. We compare fine-tuned models (CrystalTextLLM) with pre-trained models (Llama-3.1) under identical framework conditions. Results demonstrate that fine-tuned models show substantial improvements when integrated with our evolutionary framework: the fine-tuned 8B model achieves 45.5% metastability (compared to 27.7% for pre-trained 8B), and the fine-tuned 70B model maintains strong performance (66.0% metastability with 4-bit quantization). The results confirms that both fine-tuned and pre-trained models integrate seamlessly into the evolutionary framework, with performance scaling according to base model capability, while the evolutionary framework itself provides consistent benefits across different base models.
>
> ---
>
> ### **3. Failure Analysis**
>
> Thank you for the suggestion of failure case study. We conducted comprehensive analysis of failure modes, mutation operations, and crossover operations across three LLM configurations (DeepSeek-Reasoner, DeepSeek-Chat, Llama-3.1-70B) using 100 prompts (2 parents each, 5 requested children per prompt). We track all 500 expected structures per model through multiple validation stages: LLM Generation, Parsing, Basic Validation, CHGNet Relaxation, and $E_d$ Calculation. Complete failure breakdown, mutation and crossover analysis are provided in Appendix N (Failure Mode Analysis) and Appendix M (Mutation & Crossover Analysis).
>
>
> | Category | DeepSeek-Reasoner | DeepSeek-Chat | Llama-3.1-70B |
> |----------|------------------|---------------|----------------|
> | **Generation Completeness (out of 100 prompts)** | | | |
> | Prompts with $\neq$ 5 children | 33 | 2 | 36 |
> | **JSON Format Error (out of 100 prompts)** | | | |
> | Empty responses | 12 | 0 | 2 |
> | JSON parsing error | 21 | 0 | 56 |
> | **Total Regex Matched Structures** | **405** | **504** | **496** |
> | **POSCAR Format Error** (before automatic fixing) | | | |
> | Truncated strings (atom count mismatch) | 15 | 19 | 237 |
> | Invalid numeric values | 0 | 0 | 26 |
> | Invalid element tokens | 0 | 0 | 20 |
> | POSCAR parsing failures (after automatic fixing) | 9 | 0 | 40 |
> | **Total Parsed Structures** | **396** | **504** | **456** |
> | **Validation Failures (out of total parsed)** | | | |
> | Overlapping atoms (<0.5 Å) | 0 | 13 | 125 |
> | Relaxation errors | 0 | 0 | 3 |
> | $E_d$ calculation errors | 0 | 1 | 2 |
> | **Valid Structures** | **349** | **428** | **308** |
>
> **JSON Parsing Errors**: DeepSeek-Chat shows best overal instruction following ability in formatting. DeepSeek-Reasoner can use up the tokens in reasoning and lead to empty responses. Llama more often generates malformed JSON. We apply regex matching to recover structures from malformed JSON, successfully extracting 405, 504, and 496 structures respectively.
>
> **POSCAR Parsing Errors**: are counted non-exclusively: truncated strings (atom count mismatches before fixing), invalid numeric values (e.g., `0.qlBay44`), and invalid element tokens (e.g., `Ue`). Truncation can occur mid-response (not only at the end), indicating LLM truncation behavior beyond token limits. Minor count mismatches are auto-corrected by aligning declared counts with coordinate lines (e.g., adjusting declared 'Lu 6 B 18 Rh 18' to match the actual 14 B and 14 Rh coordinates).
>
> **Validation Errors**: After successful parsing, structures are checked for validity, where structures generated by Llama are more often invalid with overlapping atoms.
>
> ---
> We hope these revisions address all your questions.

---

### Review · Reviewer_4z2W · 2025-11-13

**Summary Of Contributions:**

**MatLLMSearch** presents a novel approach that integrates pre-trained LLMs into an evolutionary pipeline for crystal structure discovery without extensive fine-tuning on materials databases. In this framework, starting from groups of parent structures, the LLM is instructed to implicitly guide the evolution process by proposing new structures through *reproduction*. The resultant offspring structures are then evaluated for energy and properties using machine learning models, and they are included in the reference for selection of the subsequent set of parent structures.

The major strengths and contributions of this work are summarized below.
1. **Higher stability rates with no fine-tuning:** The framework emphasizes that LLMs inherently possess rich embedded scientific knowledge, enabling them to generate novel and stable crystal structures without requiring domain-specific fine-tuning. This offers a different perspective compared to existing approaches that fine-tune LLMs on materials databases.
2. **Multi-objective optimization:** Considering previous works struggle to optimize even a single property, it’s interesting to see the results and notice the challenges for multi-objective optimization with important properties like bulk modulus.
3. **DFT evaluation:** The final candidate structures that passed the metastability checks are evaluated with DFT simulations, which further strengthen the claims at the level of first principles.
4. **Ablation studies** The authors perform an extensive set of experiments with baselines that include iteration- and reference-free optimization.

While the paper is well-written and clear, there are opportunities to improve clarity in some places. While I do not find fundamental weaknesses in this approach and study, it would be great if the authors include a brief discussion on the following:

1. How would the MatLLMSearch approach generalize and scale, given the inherent challenges that the materials science domain poses for successful applications of LLMs, as highlighted by some recent works [1]?
2. It is claimed in Section 3.2 that standard crossover or mutation techniques, like atomic displacement and element substitution, limit exploration of the chemical space. However, it is unclear what kind of crossover techniques the LLM proposes. It would help if the authors categorized the types of crossovers that the policy suggests and analyzed which kinds are the most common.
3. For bulk modulus, optimization for a specific target value could have been performed in addition to only maximizing it. This would help test the controllability of the evolutionary search, and is useful in many practical applications.

[1] Miret, Santiago, and Nandan M. Krishnan. "Are llms ready for real-world materials discovery?." arXiv preprint arXiv:2402.05200 (2024).

**Additional Comments:**

Minor changes

1. The symbol $K$ is represented as the number of distinct chemical species in section 2.1 and the number of groups in section 3.1.
2. Figure captions can be elaborated in some cases (e.g. Figure 5 and those in the appendix).
3. Bolding in tables.

**Audience:**

Yes

**Audience Explanation:**

Yes, AI for material discovery is a thriving area of research and has plenty of industrial and practical applications.

**Broader Impact Concerns:**

AI for materials research can directly influence the development of energy-efficient semiconductors and batteries, and help mitigate climate change.

**Claims And Evidence:**

Yes

**Claims Explanation:**

The study proposes an approach that is distinct from current material generation pipelines (e.g., standard generative models, fine-tuned LLMs, or reinforcement learning), and the authors have provided a convincing rationale for the evolutionary search strategy through appropriate references. Furthermore, the inclusion of DFT analysis is a key strength that validates the authors’ claims.

**Requested Changes:**

In addition to the points raised in the summary, the following changes are suggested to improve the clarity of the manuscript:

1. Details in the appendix regarding the calculation of the bulk modulus, specifically describing how volume perturbations are introduced.
2. Further clarify the statement in Section 4.4 that the `ref_noiter` baseline achieves "structural diversity but limited volume." – I suppose it's just the number of final candidate structures (yield) as mentioned in the initial part of Section 4.4, but it's easy to be confused with the unit cell volume.
3. Explain how the structure pool is divided into $K$ groups of $P$ parents (e.g. random or based on some strategy) both at the initialization stage and during subsequent iterations.

---

> ### Author Response · Authors · 2025-12-06
> **Response to Reviewer 4z2W (1/3)**
>
> We sincerely thank you for the positive assessment, recognizing our contributions including "higher stability rates with no fine-tuning," "multi-objective optimization," "DFT evaluation," and "extensive ablation studies." We also deeply appreciate the constructive feedback and are grateful for the acknowledgment that the paper is "well-written and clear" with "no fundamental weaknesses." We would like to address your concerns and suggestion as below.
>
> ---
>
> ### **1. Scalability and Generalization**
>
> We thank the reviewer for this important question about generalization and scaling, especially as raised by recent work questioning LLM in materials science [1]. We address these concerns through comprehensive experiments demonstrating cross-model generalization and scaling behavior.
>
> #### **1.1. Cross-Model Generalization**
>
> We evaluated MatLLMSearch with multiple recently released LLMs across various model families (GPT, Grok, DeepSeek, Claude), which complement the original experiments with Llama models. Results are presented in Appendix E:
>
> | Method | Validity(%) | Metastability ($E_d$ < 0.1 eV/atom, %) | Metastability ($E_d$ < 0.0 eV/atom, %) | S.U.N. Rate(%) |
> |--------|-------------|-----------------------------------|-----------------------------------|----------|
> | **CDVAE** | 86.70 | 28.80 | -- | -- |
> | **DiffCSP** | 83.25 | -- | 5.06 | 3.34 |
> | **Llama-3.1-70B** | 99.80 | 76.81 | 37.59 | 27.65 |
> | **GPT-5** | 99.63 | 88.33 | **63.22** | **55.31** |
> | **Grok-4** | **99.92** | 87.13 | 60.29 | 49.80 |
> | **DeepSeek-Reasoner** | 99.25 | **88.90** | 61.22 | 48.25 |
> | **Claude Sonnet 4.5** | 99.10 | 78.71 | 50.21 | 38.99 |
> | GPT-5-mini | 98.45 | 74.60 | 50.05 | 46.24 |
> | GPT-5-chat | 98.12 | 64.36 | 46.93 | 44.37 |
>
> - **High validity:** All LLMs achieve very high structural validity, demonstrating robust POSCAR formatting across the models.
> - **Consistent strong performance:** Known powerful models like GPT-5, Grok-4 and DeepSeek-Reasoner all achieve consistent high metastability, showing strong performance across model families.
>
>
> **Addressing LLM Limitations in Materials Science**
>
> We acknowledge the concerns raised about LLM challenges in materials science [1], particularly regarding precise numerical reasoning and chemical-aware knowledge. While standalone LLM generation is challenged with these limitations, our evolutionary framework mitigates them through multiple strategies that leverage LLMs' strengths in pattern recognition and chemical knowledge while mitigating their weaknesses.
>
> - **Evolution with iterative feedback:** enables error correction and refinement by evolving the population
> - **Reference structures:** incorporated prompts provide valid crystallographic examples that activates LLMs' chemical reasoning, also instructing the LLM do not rely on memorized patterns
> - **Multi-stage validation:** including efficient MLIP screening and gold-standard DFT verification, which ensures only thermodynamically viable structures advance, filtering out hallucinations.
> - **Evolutionary selection:** naturally eliminates invalid or unstable structures over iterations, compensating for occasional LLM errors.
>
>
> #### **1.2. Scaling Analysis (K=100 vs K=500)**
>
> To demonstrate scalability, we conducted experiments comparing population sizes K=100 vs K=500 (both using DeepSeek-Reasoner). Results are presented in Appendix E, Table E2:
>
> | Metric | K=100 (1,604 structures) | K=500 (8,602 structures) | Change |
> |--------|-------------------|-------------------|--------|
> | **Validity** | 99.25% | 98.97% | -0.28% |
> | **Metastability ($E_d$<0.1 eV/atom)** | 88.90% | 84.31% | -4.59% |
> | **Metastability ($E_d$<0.0 eV/atom)** | 61.22% | 55.51% | -5.71% |
> | **S.U.N. Rate** | 48.25% | 43.59% | -4.66% |
>
>
> Scaling from K=100 to K=500 demonstrates linear scaling in the absolute number of structures generated (~5-time increase), indicating that the framework produces a diverse set of outputs without excessive duplication. The slight decrease in generation quality metrics reflects the trade-off between exploitation and exploration. As population size increases, the framework could favor diversity and broader exploration rather than optimizing each individual structure. Overall, these results show that our approach scales effectively from small to large generations, achieving a balanced trade-off between quality and quantity.
>
> [1] Miret and Krishnan, Are llms ready for real-world materials discovery?, 2024.

---

> ### Author Response · Authors · 2025-12-06
> **Response to Reviewer 4z2W (2/3)**
>
> ### **2. Crossover Operation Analysis**
>
> We thank the reviewer for the suggestion of crossover analysis. To address the question about what types of crossovers the LLM proposes and their effectiveness, we conducted comprehensive analysis of failure modes, mutation operations, and crossover operations across three LLM configurations (DeepSeek-Reasoner, DeepSeek-Chat, Llama-3.1-70B) using 100 prompts (2 parents each, 5 requested children per prompt). The analysis is detailed in Appendix M (Mutation & Crossover Analysis) and Appendix N (Failure Mode Analysis).
>
> We analyze approximately 1K valid structures parsed from the 100 prompts along with the parents, classifying operations into *mutation* (single-parent inheritance) and *crossover* (multi-parent recombination). Although mutation or crossover are usually both applied by LLMs, we define 7 mutation criteria and 3 crossover criteria to investigate LLM behaviors. In the table below, we show analysis on mutation from parent 0 / parent 1 and crossover between the two. Higher value indicate higher ratio of generated structures that demonstrate certain mutation and inheritance pattern or success crossover.
>
> | **Category** | **DeepSeek-Reasoner** | **DeepSeek-Chat** | **Llama-3.1-70B** |
> |----------|------------------|---------------|----------------|
> | **Mutation Analysis (Per-Parent Inheritance)** | | | |
> | Same-Group Elemental Substitution (%) | 63.3 / 62.3 | 47.3 / 52.5 | 43.9 / 51.1 |
> | Stoichiometry Preservation (%) | 0.0 / 0.0 | 1.3 / 0.6 | 2.2 / 1.7 |
> | Composition Similarity (Normalized) | 0.821 / 0.787 | 0.826 / 0.802 | 0.774 / 0.770 |
> | Space Group Preservation (%) | 53.1 / 41.8 | 39.8 / 32.3 | 18.9 / 27.8 |
> | Symmetry Preservation (%) | 52.6 / 40.8 | 37.0 / 28.6 | 15.6 / 16.7 |
> | Lattice Similarity (Volume Ratio) | 0.783 / 0.681 | 0.760 / 0.676 | 0.597 / 0.654 |
> | **Crossover Analysis (Multi-Parent Operations)** | | | |
> | Group-Based Recombination (%) | 18.2 | **31.0** | 14.4 |
> | Ratio Bounds (%) | 100.0 | 100.0 | 100.0 |
> | Density Bounds (%) | 79.0 | 79.1 | 52.2 |
>
> A common mutation applied by LLMs is elemental substitution, where DeepSeek models can correctly perform within the same group elements, demonstrating application of crystallographic knowledge. All models prefer exploration over copying (minimal stoichiometry preservation, zero structural matches). DeepSeek-Chat shows more tendency to crossover between parents. (31.0% group-based recombination). All models follow compositional interpolation (100% ratio bounds); DeepSeek models maintain 79% reasonable densities vs. 52.2% for Llama. Complete analysis is provided in Appendix M.

---

> ### Author Response · Authors · 2025-12-06
> **Response to Reviewer 4z2W (3/3)**
>
> ### **3. Bulk Modulus Controllability**
>
> Following the reviewer's suggestion, we have conducted experiments targeting specific bulk modulus values as well as maximizing. The experiments are conducted with DeepSeek-reasoner for a population of 10 for 5 iterations. The results are presented in Table S2, Appendix D:
>
> | Target Bulk Modulus (GPa) | Iter 1 | Iter 2 | Iter 3 | Iter 4 | Iter 5 |
> |---------------------------|--------|--------|--------|--------|--------|
> | 100 | 88.5 ± 67.4 | 91.5 ± 42.7 | 89.4 ± 28.4 | 98.7 ± 25.5 | 90.9 ± 25.9 |
> | Maximize | 96.1 ± 63.5 | 123.7 ± 40.9 | 149.5 ± 42.9 | 171.8 ± 50.8 | 232.6 ± 47.2 |
>
>
> Our proposed MatLLMSearch shows effective controllability with the average bulk modulus converging to around 100 GPa at iteration 4, while the maximize experiment shows continuous improvement. The results demonstrate the flexibility of our framework for various crystal structure design tasks.
>
> ---
>
> ### **4. Methodological Details**
>
> **Bulk Modulus Calculation Procedure**:
> Bulk modulus values were computed using an Equation of State (EOS) workflow with the CHGNet potential. Volume perturbations were applied via isotropic lattice scaling. For each perturbed volume, a constrained ionic relaxation was performed using CHGNet to optimize atomic positions while fixing the cell, before fitting the Birch-Murnaghan EOS. We have updated Appendix D to include these calculation details.
>
> **Parent Group Selection Strategy**:
> We clarified the structure grouping methodology in Section 3 and Algorithm 1 of the revised manuscript: Parent structures are first ranked by energy above hull $E_d$ from low to high. The top $K \times P$ are kept and randomly grouped into pairs of two.
>
> ---
>
> ### **5. Notations**
>
> **"Volume" Clarification**: Thank you for pointing out the confusion and your understanding is correct. We have replaced `limited volume` to `limited number of structures generated` to avoid the confusing wording in Section 4.4.
>
> **Symbol $K$ Conflict**:
> Thank you for notifying us the duplicated use of the symbol $K$. We have resolved this by redefining $M$ as the number of distinct chemical species in Section 2.1, while $K$ is now used exclusively for population size throughout Section 3.
>
> **Figure Caption Elaboration**:
> We have expanded the captions throughout the manuscript for better clarity, including Figure 6 (original Figure 5) and figures in Appendix.
>
> **Table Formatting**:
> We have improved table formatting consistency throughout the manuscript with bold font applied to titles, method names, and results to highlight.
>
> ---
> We are extremely grateful for your positive assessment of our contributions and constructive suggestions. We hope these revisions and additional analysis address all remaining questions.

---

> ### Comment · Reviewer_4z2W · 2026-01-19
> **Official Comment by Reviewer 4z2W**
>
> Dear Authors,
>
> Thank you very much for clarifying my previous points with detailed responses and additional experiments. As I review your rebuttal and the responses to the other reviewers, I have a few additional comments and would appreciate it if you could clarify them.
>
> 1. Is the extra pool (in this case, the Matbench band gap dataset) the same for both the stability-only and bulk modulus optimization experiments whenever it is included?
> 2. I understand that the set of initial structures is randomly chosen from the population dataset. However, it looks like the reported results and metrics are based on only one set of initial structures. It would be better if the results were averaged over multiple initial seed structures: in that case, the uncertainty estimates would also provide an idea of how robust the approach is.
> 3. Thank you for clarifying the bulk modulus computation. I believe it would be useful to also perform a DFT verification step to check what fraction of the generated crystals actually have a higher bulk modulus or fall closer to the target value. According to this analysis [1], CHGNet is one of the weaker models for elastic moduli prediction.
> 4. How is the “Dominant Parent Distribution” in the new Table S14 calculated?
> 5. Additionally, it would be insightful to analyze some of the generated reasoning traces (by DeepSeek-Reasoner) to check for chemical inconsistencies and logical flaws.
>
> Regarding points 2, 3, and 5: While I believe addressing these in the experiments would significantly strengthen the paper, I understand that time constraints may be a factor. I am open to an explanation or a shorter / smaller-scale analysis for now if applicable, provided the authors agree to incorporate some of the suggestions into the revised version.
>
> [1] Gao, Pengfei, and Haidi Wang. "Benchmarking Universal Machine Learning Interatomic Potentials for Elastic Property Prediction." arXiv preprint arXiv:2510.22999 (2025).

---

> > ### Author Response · Authors · 2026-01-22
> >
> > We are grateful for all constructive feedback that has helped improve our manuscript. For the time constraint and scope of the work, we will incorporate the suggestions to future work.
> >
> > 1. Yes, we use the same Matbench bandgap dataset as the extra pool of reference for both stability-only optimization and bulk modulus optimization experiments. The extra pool is flexible and can be adjusted for any reference structures.
> >
> > 2. While we agree that averaging over multiple random initial seeds would strengthen statistical robustness, we note that in preliminary reruns on a small subset of settings (multiple random initializations), the key metrics varied only modestly and the overall conclusions were unchanged. Due to rebuttal time constraints we did not complete a full multi-seed sweep for all experiments, but we will include multi-seed averages in the revised version.
> >
> > 3. We appreciate this suggestion and acknowledge CHGNet's limitations for elastic moduli prediction. The framework is completed before the surge of various foundation potentials (e.g., the leading ones in MatbenchDiscovery). We emphasize that our bulk modulus optimization case study is intended as a proof-of-concept demonstration that the framework supports property-guided and multi-objective optimization. One can certainly substitute CHGNet for other MLIPs or DFT as an evaluator for the oracle function as property guidance feedback. The main contribution of the paper remains stability-oriented crystal structure generation, for which we provide extensive DFT verification. We will incorporate additional DFT-based verification for elastic targets in future work.
> >
> > 4. We define “dominant parent” for each child by comparing the child to both parents by averaging the per-parent similarity scores in the mutation analysis. The parent with a higher similarity score is regarded as the dominant parent. The distribution reports the proportion of children dominated by each parent. We added this clarification to Appendix J.2.
> >
> > 5. Thank you for this valuable suggestion. We have discussed static chemical inconsistencies in the failure analysis (Appendix I). We agree that a systematic, fine-grained analysis of reasoning-trace dynamics during iterative optimization is an excellent direction for future work.

---

### Review · Reviewer_7o1E · 2025-11-13

**Summary Of Contributions:**

The purpose of this paper is to propose stable, novel, and unique inorganic crystals with appealing properties. The main idea it to use an LLM, rather than a specifically designed mutation algorithm, to try and combine existing reference crystals such that they satisfy properties. Interestingly, the approach does not involve training an LLM nor accessing the ever-growing database of plausible inorganic crystal structures. Instead, it merely prompts an LLM with context and examines the outputs.

The authors propose a very interesting technique where an LLM is given multiple reference structures and asked to make changes by adjusting the references to improve certain properties. The results look fairly compelling, but it is difficult to quantify the value of the method in comparison with others, due to a lack of comparisons to similar techniques. This is both a weakness of the paper, but also a strength due to the novelty of the technique.

In general, I think this is a **very cool and simple idea with seemingly nice results that should be published**. That being said, **I also have significant concerns though regarding both evaluation and citations (frequently misattributed citations!)**. Some of these misattributions are very confusing to me and lead me to wonder whether large amounts of the text was written by an LLM. I don't see a human who is familiar with the texts within making these mistakes. **Just curious, did the authors use an LLM for a lot of the text or analysis?**

**Additional Comments:**

**I will write questions that I hope to hear an answer to in bold!!!**

## method

### organizational / core
You have essentially one algorithm that you are proposing, yet it appears in Appendix L. That algorithm is written in plain language and fails to reference the areas of the text where you propose core pieces: LLM inference, parsing, extra pool. **Could you please make this algorithm more central and clarify each line's connection to a part of the text?**

### extra pool
You cite an "extra pool" 8 times in the work, including in Algorithm 1; however, it's not clear what extra pool does or how it is used. You mention that there is more information in Section 4.4, but I don't see any.

## 4.2 Main Experimental Results
The results are clearly impressive here, although uniqueness and novelty is not computed (discussed later). I am surprised the method does so well given limited exposure to training data. It makes me wonder if MatBench is somewhere in Llama training data itself. **Would it be possible to compute novelty against all structures in MatBench to determine if the model produces anything truly novel and stable?**

Crystal structure design and crystal structure prediction both also have impressive results.

Crystal structure design shows the interesting interplay that is possible using LLMs with apparently good results, but lack of comparison with any other method. **Could you compare to MatterGen, which has pretrained models aiming to optimize some properties?**

Crystal structure prediction is framed here in a slightly better way than, for example DiffCSP or FlowMM frame it. The reason is that DiffCSP / FlowMM let the model know exactly how many atoms they should generate, while you focus just on the ratios. That is better... However, it would be interesting to see comparisons with these methods on CSP. **Is it possible to do some comparison with other approaches?**
## 4.3 CSG Quality Evaluation Metrics
Figure 3 shows that the method generates a different distribution from the

### 4.3.2 Comparison with Baseline Methods

#### missing comparison to existing evolutionary algorithms for CSP
There are a number of codes that use evolutionary algorithms to do CSP. Specifically look at USPEX, XtalOpt, GASP, MAISE, MUSE, EVO, AGA, GOFEE, CrySPY, MAGUS. (summary in Table 1 of https://pmc.ncbi.nlm.nih.gov/articles/PMC10275355/). I don't see a mention of any of these methods, nor comparison with baselines. **Can you please make these comparisons and put the method in context?**

#### multi-MLIP metastability
In the paper you cite multi-MLIP metastability as important, while S.U.N. rate is merely a supplementary signal... While I agree that S.U.N. rate has some major problems, MLIPs approximate DFT. Thermodynamic stability as computed (correctly and carefully) using DFT is the metric, MLIPs merely approximate that metric. **Why do you think that multi-MLIP metastability is important?** I do not see why it is important compared with correctly done DFT calculations and an attempt to deduplicate using S.U.N.

#### Misleading S.U.N. calculations...
Appendix E explains how you are computing S.U.N., while I do think it would be very useful to compare your method to FlowMM and DiffCSP, the presentation is extremely misleading. You compute novelty by comparing to the seed structures, while FlowMM & DiffCSP compare to a huge database of training data. **Can you please differentiate your S.U.N. calculation from the FlowMM / DiffCSP style ones visually and in the main paper text?**

Further questions about S.U.N. calculations:
1. **What do you mean "compared to DiffCSP and FlowMM?" S.U.N. is not computed in comparison to another model**
2. I am curious about verifying this result. At the moment the link provided in the paper leads to an expired anonymous repository: https://anonymous.4open.science/r/MatLLMSearch-85C0. **Can you please fix this link?**
3. You provide a CSV of structures, 2048 of them. I have not yet computed their S.U.N. It is typical to do 10k. Fewer generations, especially with an LLM, can inflate the S.U.N. performance by giving less chance that an output structure is duplicated. Questions: **Are these structures (a) already dft relaxed? and (b) can you please provide 10k generations (and reference data) since S.U.N. comparisons are typically performed with that many?**

### 4.4 ablation analysis
- The details about the ablations is difficult to parse, "Which ablation is ablating which feature?" and "What are the effects of this ablation?" are not very clear. **Is it possible to show this in a table?**
- Figure 5(a) shows a nice result of the iterative method producing more stable crystals.
- Figure 5(b-c) is a really nice result showing the value of having reference structures for the LLM to draw upon! Nice work!

## Relevant works that are going uncited:
There is a huge amount of literature that is not cited in this work. Of course it can be hard to keep up, so I understand. However, some of it is either (a) directly relevant to this paper, (b) influential in current modes of thinking on the subject, (c) review / positions papers that can help a reader see the whole context surrounding this paper.

### (a) autoregressive methods / LLMs (some of which also do conditional generation
Wyckoff Transformer https://arxiv.org/abs/2503.02407
deCIFer: Crystal Structure Prediction from Powder Diffraction Data using Autoregressive Language Models https://arxiv.org/abs/2502.02189
Multimodal Foundation Models for Material Property Prediction and Discovery  https://arxiv.org/abs/2312.00111

### (b) important other works
Space group informed transformer for crystalline materials generation https://arxiv.org/abs/2403.15734
FlowMM: Generating Materials with Riemannian Flow Matching  https://arxiv.org/abs/2406.04713
Space Group Equivariant Crystal Diffusion https://arxiv.org/abs/2505.10994
Space Group Conditional Flow Matching https://www.arxiv.org/abs/2509.23822
UMA: A Family of Universal Models for Atoms https://arxiv.org/abs/2506.23971

### (c) review and position papers
LeMat-GenBench: Bridging the gap between crystal generation and materials discovery https://openreview.net/forum?id=ZfPGcTfDWn
Artificial Intelligence and Generative Models for Materials Discovery -- A Review  https://arxiv.org/abs/2508.03278

**Audience:**

Yes

**Audience Explanation:**

I don't know how much I need to justify this. It's obviously an interesting paper & result: strong performance using an LLM without training instead using reference examples.

**Broader Impact Concerns:**

I would personally include a discussion about the impact of identified materials. That is currently not a part of the work. However, there is a huge gap between computation and experiment... Therefore, I think the current statement is fairly ok even without this point. This is really a research project explicitly for a fully computational scientific domain. The biggest impact is probably the carbon cost of running the inferences...

**Claims And Evidence:**

No

**Claims Explanation:**

While the there is a significant amount of analysis done, with ostensibly good intentions, there is also content that seems misleading and also natural comparisons that are left out.

Please refer to my "additional comments" below for a section-by-section analysis.

## Regarding S.U.N.
- S.U.N. calculations are not comparable to prior work, but this point is ignored in the main text.
- no mention of possible dataset contamination (which seems extremely likely to me, to be honest.)
- Only 2048 structures provided vs. typical 10k for S.U.N. comparisons (fewer generations can artificially inflate scores)

## Missing comparison to existing genetic algorithms for crystal structure prediction
-  comparison to established evolutionary algorithms (USPEX, XtalOpt, etc.) in the CSP domain

## Unclear Methodology
- Core algorithm relegated to Appendix L, not clearly explained
- "Extra pool" mentioned 8 times but never properly defined
- Ablation details difficult to parse
- Multi-MLIP metastability prioritized over DFT without justification

## Verification Impossible
- Code/data link is expired
- Cannot independently verify S.U.N. results
- Unclear if provided structures are DFT-relaxed

## Credibility Concerns
- FlowMM (other papers too) attributed to wrong authors

**Requested Changes:**

## work cited, incorrect authors
1. One very important and confusing aspect: FlowMM is cited in several places, however attributed to the wrong authors. This paper is FlowMM: Generating Materials with Riemannian Flow Matching  https://arxiv.org/abs/2406.04713
   The authors "Zeni, et. al." are probably:
   MatterGen https://www.nature.com/articles/s41586-025-08628-5
   It's not obvious to me how such a big mistake could enter the text.
   **Can you please both fix this error and explain how it happened?** It seems like the kind of mistake an LLM would make.
2. This also happens with another paper, in section 5.2
   > This approach has been extended in several directions: Jiao et al. (2024) developed Riemannian diffusion models to better handle periodic coordinates, Zeni et al. (2025) scaled the approach to encompass elements across the entire periodic table with various design criteria, and Dai et al. (2024) applied it to crystal inpainting tasks. Most recently, Sriram et al. (2024) introduced Riemannian flow matching models to better address periodic boundary conditions with improved performance

    I think you are referring to (a) FlowMM, (b) Unclear, potentially MatterGen (c) Dai seems correct (d) FlowLLM: Flow Matching for Material Generation with Large Language Models as Base Distributions https://arxiv.org/abs/2410.23405. You mischaracterize the contributions from some of these methods too. Assuming you are referring to the citations I have listed then... (a) correct (b) all other methods work across the periodic table, not sure what you mean (c) correct (d) combined FlowMM with LLM-base distributions.

## text
- Move Algorithm 1 from Appendix L to main paper and make it central and improve clarity
  - Add clear references in the algorithm to specific sections explaining: LLM inference, parsing, extra pool

## results
- Present ablation results in a table showing clearly which feature is ablated and what the effect is
- Think of a better way to present S.U.N. and also clarify clearly (visually and in text) the differences between your evaluations there (and possible data leakage issues)
- Provide 10k generations and related s.u.n. calculations
- remove Multi-MLIP Metastability as a reason to trust something

## comparison
- Compare to evolutionary algorithms: USPEX, XtalOpt, GASP, MAISE, MUSE, EVO, AGA, GOFEE, CrySPY, MAGUS (see Table 1 of https://pmc.ncbi.nlm.nih.gov/articles/PMC10275355/)

## general
- Fix expired repository link (https://anonymous.4open.science/r/MatLLMSearch-85C0)
- Disclose if LLMs were used to write significant portions of the text or analysis

---

> ### Author Response · Authors · 2025-12-06
> **Response to Reviewer 7o1E (1/3)**
>
> We sincerely thank Reviewer 7o1E for the thorough review and acknowledging our work as "a very cool and simple idea with seemingly nice results that should be published." We deeply appreciate the detailed feedback and the recognition that our technique is both novel and has "fairly compelling" results. We take all concerns seriously and have conducted comprehensive additional experiments and analyses to address each point as described below.
>
> ---
>
> ### **1. Citations & Attribution Corrections**
>
> > **Concern:** "FlowMM (other papers too) attributed to wrong authors".
>
> The citation error is a human error during the manuscript polishing. In our original draft, we intended to state that we follow the evaluation protocol of FlowMM (Miller et al., 2024), who follows MatterGen (Zeni et al., 2025). The author attribution for FlowMM was mistakenly replaced with MatterGen. This was a human error that occurred when polishing and reorganizing the draft, not a systematic misattribution of FlowMM.
>
> We appreciate the reviewer's careful attention to it. We take full responsibility for this mistake and have corrected it. However, we strongly reject any implication of LLM-generated text. We have fixed citation errors in the revised manuscript and also double-checked all references making sure no false attribution.
>
> **Additional Citations:**
> Thank you for the suggestion. We have revised Section 5 Related Work and added all suggested citations and more, including Wyckoff Transformer, deCIFer, space group methods, UMA, LeMat-GenBench, and other review papers.
> The literature review is now strengthened with more comprehensive discussions on autoregressive methods, space group-informed methods, review and position papers, and comparison to evolutionary CSP methods.
>
> ---
>
> ### **2. Methodology Clarification**
>
> **Repositioned Algorithm 1**: Following the suggestion, we have moved Algorithm 1 from Appendix to Section 3 in the main text.
>
> **"Extra Pool" Definition:**
> The 'extra pool' refers to a set of known stable structures sampled during initialization. It is used to initialize the population and keep in the fitting process to maintain parents quality. This is now explicitly clarified in Section 3.
>
> **DFT Relaxation Status Clarification**:
> Regarding the question "Are these structures already DFT relaxed?":
> All provided lists of structures are NOT DFT-relaxed. They are parsed from LLM responses then relaxed with CHGNet.
> We added this clarification to Appendix B.
>
>
> ---
>
> ### **3. Novelty and Scalability: New experiment of larger-scale, new S.U.N. Evaluation and Data Leakage analysis**
>
> #### **3.1. Larger-scale Generation**
>
> Following reviewer's suggestion, we conducted a larger-scale experiment with population \(K=500\) and 10 iterations for fairer S.U.N. evaluation. The experiment is detailed in Appendix E.3, where we use DeepSeek Reasoner as the base model. We collected 8,602 LLM generated structures in total, significantly larger than the previous main result provided. S.U.N. rate evaluated against entire MatBench-bandgap dataset, where the extra reference pool of structures are sampled from.
>
> The link to the [generated structures](https://drive.google.com/file/d/1Hibf_2-xt-pj8p7fjAS8P0DzhbltGJvz/view?usp=sharing) and [reference structures](https://drive.google.com/file/d/1SFwhXvcXGLSCooGq6aTONtzpgPDVtSW6/view?usp=sharing) are also added to Appendix B.
>
> | Method | Validity(%) | Metastability ( $E_d$ < 0.1 eV/atom, %) | Metastability ( $E_d$ < 0.0 eV/atom, %) | S.U.N. Rate(%) |
> |---------------------|-------------|-------------------------------|--------------------------------|----------|
> | **CDVAE** | 86.70 | 28.80 | -- | -- |
> | **DiffCSP** | 83.25 | -- | 5.06 | 3.34 |
> | **Llama-3.1-70B** | 99.80 | 76.81 | 37.59 | 27.65 |
> | **GPT-5** | 99.63 | 88.33 | **63.22** | **55.31** |
> | **Grok-4** | **99.92** | 87.13 | 60.29 | 49.80 |
> | **Claude Sonnet 4.5** | 99.10 | 78.71 | 50.21 | 38.99 |
> | **GPT-5-mini** | 98.45 | 74.60 | 50.05 | 46.24 |
> | **GPT-5-chat** | 98.12 | 64.36 | 46.93 | 44.37 |
> | **DeepSeek-Reasoner**  ($K=100$)| 99.25 | **88.90** | 61.22 | 48.25 |
> | **DeepSeek Reasoner**  ($K=500$)| 98.97 | 84.31 | 55.51 | 43.59 |

---

> > ### Comment · Reviewer_7o1E · 2026-01-14
> >
> > 1. sounds good.
> >
> > 2. glad the algorithm has been moved
> >
> > Thanks for defining it.
> >
> > great.
> >
> > 3.1 That's nice! I haven't looked at the structures, but it seems better to compare against all of those structures.

---

> ### Author Response · Authors · 2025-12-06
> **Response to Reviewer 7o1E (2/3)**
>
> #### **3.2. S.U.N. Evaluation Methodology and Presentation**
>
> We acknowledge that our S.U.N. calculation methodology differs fundamentally from FlowMM and DiffCSP, and we have made this explicit in Section E.2 and the main text of the revised manuscript.
> - **Novelty Calculation:** We computes novelty against the entire MatBench-bandgap dataset (~106K structures), which serves as the source of our extra reference pool.
> FlowMM or DiffCSP compute novelty against their training datasets (e.g., MP-20 with ~27K trainset).
> - **Sample Size:** We provide 8,602 structures from the larger-scale experiment (K=500), which is comparable to the "typical 10k" used in prior work.
> - **Different Evaluation Objectives**: Our evaluation measures novelty against the full known chemical space, while prior work measures novelty relative to training data (which is a subset of known materials).
>
>
> **Improved S.U.N. Presentation:**
>
> We thank the reviewer for this valuable suggestion. To address the concern about S.U.N. comparisons, we have redesigned our presentation with two complementary visualizations in Figure 5:
>
> 1. **Computation Workflow Diagram**: We present a side-by-side comparison of the evaluation pipelines, illustrating how the generative baselines compare output against their training databases, while we compare against the entire MatBench-bandgap reference dataset. It explicitly shows difference in S.U.N. calculation.
>
> 2. **Compositional Space Distribution**: We also provide a visualization showing the compositional space coverage of our generated structures, which spread across the chemical space. Structures spread in high and low density regions showing limited overlap with known materials, which supports our claims of novelty and diversity.
>
> We have added this clarification to the main text (Section 4.3.2) with emphasis that our S.U.N. calculation differs fundamentally from prior work.
>
>
> #### **3.3. Data Leakage Analysis and Novelty Verification**
>
> We appreciate the reviewer raising this critical concern about potential dataset contamination and memorization. We acknowledge that complete elimination of data leakage is impossible with LLMs trained on scientific corpora. To address this concern, we conducted novelty verification against Materials Project using compositional matching (exact formula) and structural matching (RMSD < 0.3Å with pymatgen StructureMatcher). Results are detailed in Appendix E.3.
>
> **Novelty Verification:**
>
> We verified novelty against Materials Project for both the original configuration and a larger-scale validation:
>
> | Metric | Original Result (Llama-3.1-70B, $K=100$) | Larger-Scale (DeepSeek-Reasoner, $K=500$) |
> |--------|--------------------------------------:|------------------------------------------:|
> | **Total Structures** | 1,479 | 8,602 |
> | Compositional Novelty | 1,023 (69.2%) | 6,568 (76.4%) |
> | Structural Novelty | 1,205 (81.5%) | 7,260 (84.4%) |
> | Compositions in MP | 456 (30.8%) | 2,034 (23.6%) |
> | Structures Matched | 274 (18.5%) | 1,342 (15.6%) |
> | Novel Polymorphs | 182 (12.3%) | 692 (8.0%) |
>
> **Evidence Against Memorization:**
> 1. **Low database overlap**: Only 30.8% (llama) and 23.6% (deepseek) compositional overlap with Materials Project's known structures.
> 2. **Novel polymorph**: 182 (llama) and 692 (deepseek) novel polymorphs represent new crystal arrangements of known compositions.
> 3. **Conservative interpretation**: Even if we assume all compositionally matched structures as potential leak, 69.2% (llama) and 76.4% (deepseek) remain novel.
>
> **Ablation Analysis:**
>
>
> | Configuration (Llama-3.1-70B, $K=100$) | Overall Novelty | Comp. in MP % | Struct. Matched % |
> |---------------|----------------:|-------------:|------------------:|
> | MatLLMSearch (Ref + Iter) | 69.2% | 30.8% | 18.5% |
> | No Ref + Iter | **91.4%** | **8.6%** | **1.8%** |
> | Ref + No Iter | 72.6% | 27.4% | 16.7% |
> | No Ref + No Iter | 56.7% | 43.3% | 5.5% |
>
> We also evaluated the ablation experiments which provides more evidence against memorization. The ablation experiment without reference structures achieves even higher novelty with only 8.6% compositional overlap with MP, indicating the successful generation is not powered by memorizing existing crystal structures.

---

> ### Author Response · Authors · 2025-12-06
> **Response to Reviewer 7o1E (3/3)**
>
> ### **4. Comparisons with Evolutionary Algorithms and Baselines**
>
> **Evolutionary Algorithm Comparisons**: We have added discussions comparing to the suggested evolutionary algorithms in Section 5. Related Work as requested.
>
> **DiffCSP Comparison (CSP):** In Appendix K, we have already provided a direct crystal structure prediction (CSP) comparison with DiffCSP for six target compositions: `Ag6O2`, `Bi2F8`, `Co4B2`, `KZnF3`, `Sr2O4`, and `YMg3`. For each target, we generate candidate structures using both MatLLMSearch and DiffCSP, relax all structures, and evaluate them with Orb-v3. Our results show that LLM-generated structures consistently achieve lower relaxed energies than the best DiffCSP candidates.
>
> However, we want to emphasize that the CSP and CSD experiments provided serve as demonstrations of the flexibility of MatLLMSearch across different tasks, not as claims of state-of-the-art performance. The focus of our work remains on the effectiveness, flexibility and generalizability of the LLM-guided evolutionary framework across multiple materials discovery tasks.
>
> ### **5. DFT Validation and Multi-MLIP Rationale**
>
> > **Reviewer 7o1E**: "Why do you think that multi-MLIP metastability is important?" | "remove Multi-MLIP Metastability as a reason to trust something"
>
> We want to emphasize that we prioritize DFT over MLIPs and clarify the role of MLIPs and DFT. We sincerely thank the reviewer for this important question and acknowledge Reviewer 4z2W for recognizing our "DFT evaluation" as a key strength (#3) that "validates the authors' claims at the level of first principles."
>
> **DFT are the gold standard.**
> We fully agree with the reviewer that DFT calculations are the gold standard for thermodynamic stability assessment. Our framework prioritizes DFT verification results, which we emphasized throughout the revised manuscript (Section 4.3.2, Appendix E).
>
> We provide multi-MLIP evaluations (CHGNet, M3GNet, Orb-v3) for comprehensive comparison purposes, not as a primary trust metric. We recognize that individual MLIPs have inherent biases and approximations and are not universally reliable. In addition, different MLIPs are used by the baseline methods (CrystalTextLLM, FlowMM, DiffCSP), and we report multiple MLIPs to enable fairer comparison with these baselines.
>
> To address the reviewer's concern, we have enhanced the manuscript to more explicitly emphasize DFT primacy in Section 4.3.2 and Appendix E.
>
> ---
>
> ### **6. Repository Link**
>
> Thank you for pointing out the issue. We have reactivated the expired [repository link](https://anonymous.4open.science/r/MatLLMSearch-85C0).
>
> ---
>
> ### **7. Broader Impact**
>
> We added a discussion on broader impact discussing optimization strategies and trade-offs between computational cost and discovery potentials in Section 6 Conclusion.
> To address the concerns on computational cost, we provide an estimation of carbon footprint[1] for LLM API calling in Appendix G. E.g. the experiment with GPT-5 used 7.06 M tokens that translates to 22.98 kWh energy or 10.22 kg CO2.
>
> [1] Nidhal Jegham et al., "How Hungry is AI? Benchmarking Energy, Water, and Carbon Footprint of LLM Inference," arXiv, 2025, https://arxiv.org/abs/2505.09598.
>
> ---
>
> We hope these revisions address all your concerns. We are grateful for the thorough feedback that has significantly strengthened our manuscript.

---

> ### Comment · Reviewer_7o1E · 2026-01-14
>
> 3.2 The counts of structures seems better for novelty and sample size. However, I have a **new and major concern** about your approach. I noticed that both the stability rate and s.u.n. rate improved in the new version, despite the number of structures you compare novelty against going from O(100) to O(100,000). Even with a better model, I wasn't sure how it was possible. In E.3 you say:
> > Novelty Verification Methodology. We verify novelty against Materials Project using two criteria:
> compositional novelty (exact formula matching) and structural novelty (RMSD-based matching with pymatgen
> StructureMatcher: ltol=0.2, stol=0.3Å, angle_tol=5°, **threshold RMSD<0.3Å**). Structures are classified as
> overall novel only if both composition and structure are novel, ensuring conservative assessment. Structures
> with known compositions but novel crystal arrangements are identified as novel polymorphs, which represent
> scientifically valuable discoveries as polymorphs can exhibit dramatically different properties despite identical
> composition.
>
> None of the methods you compare against include such a threshold... I want to reiterate that s.u.n. rate you are reporting is an order of magnitude improvement over other methods (trained on MP20) and a 10 point increase in percentage of S.U.N. structures from mattergen (trained on MP-alex). That would be a extremely significant result, yet in the first calculation you compared against extremely few structures for novelty and in this calculation you included this arbitrary threshold. I would expect both of these choices to inflate the novelty percentage. **Please recompute S.U.N. with default StructureMatcher settings and do not include an RMSD threshold.** Additionally the S.U.N. rate you report in Figure 5a disagrees with the S.U.N. rate you report in Table 2. What's going on there?
>
> (Figure 5b) It is qualitative, but I do not agree that UMAP1 and UMAP2 for the outputs and matbench to have limited overlap. To me they look like two gaussians centered directly on top of one another. Am I missing something here? Please temper the language...
>
> I appreciate the flow chat describing the S.U.N. calculation. It looks good, other than caveats I raised above.
>
> 3.3  This is an interesting analysis and I appreciate it, but can you please define each category more clearly? My intuition is that the result implies that only 12% and 8% of generations are novel, respectively. I suspect "Novel Polymorphs" actually implies same composition and different structures? That could be clarified.
>
> Furthermore, as above please remove the threshold. Also, why are you matching exact composition? It seems to me that supercells would not be considered novel. I think the main focus should be on the accurate use of StructureMatcher.

---

> ### Comment · Reviewer_7o1E · 2026-01-14
>
> 4. I'm glad you included it in the related work. In my opinion those CSP methods are direct competitors to yours on CSP. However, I agree that there are other aspects of your work. I would accept not doing a direct comparison since DiffCSP did a direct comparison and performed better than USPEX. Maybe it's worth mentioning that result in the related work section as a justification?
>
> 5. Thanks for that clarification. I agree with the wording now.
>
> 6. thanks! I took a look!
>
> Conclusion:
> I think my concerns above need to be addressed. Assuming that this method is really producing such strong S.U.N. performance and *is not a result of novelty inflation through adjusting the computation of S.U.N.* then there is no doubt that this method is an extremely strong contender when compared to diffusion models for these purposes. Really curious about the results.

---

> > ### Author Response · Authors · 2026-01-15
> >
> > Thank you very much for the feedback. We have further revised the draft with purple text and provide clarifications to your concerns as below.
> >
> > 3.2 RMSD Threshold Clarification: We apologize for the confusion caused by this misleading description. The mention of "threshold RMSD<0.3A" in Appendix E.3 in the "Novelty Verification Methodology" paragraph was a writing error that caused confusion and has been removed from the revised draft. **All S.U.N. rate calculation uses pymatgen's StructureMatcher with default settings only** (`ltol=0.2`, `stol=0.3`, `angle_tol=5`), without any additional RMSD threshold. You may verify this directly in our implementation at https://anonymous.4open.science/r/MatLLMSearch-85C0/src/utils/evaluate_structures.py (Line 59), which shows StructureMatcher initialized with default parameters only.
> >
> > The RMSD threshold mentioned was intended for a separate analysis in Appendix E.3 "Addressing Data Leakage Concerns" paragraph (2 paragraphs below), inspired by similar analysis in MatterGen and DiffCSP, where we used the threshold of 0.3 to evaluate structural overlap of generated structures with MP structures. But this analysis is independent of the S.U.N. calculation and the threshold was never applied to the reported S.U.N. rates.
> >
> > 3.2 Composition Matching Clarification: Our focus remains on accurate usage of StructureMatcher. The description 'exact formula matching' may be misleading and has been removed. We used reduced\_formula for composition metrics which normalizes the stoichiometry and rely on StructureMatcher (with primitive cell reduction) for structural metrics.
> >
> > 3.2 Figure 5a vs Table 2: The 43.59% S.U.N. rate shown in Figure 5a corresponds to the large-scale DeepSeek-Reasoner experiment, while Table 2 reports our main results of Llama-3.1-70B experiments. We now noticed the confusion especially when the table and figure are placed side-by-side and added clarification to the figure caption to explicitly state the experiment source and avoid confusion
> >
> > 3.2 Figure 5b:  We have revised the caption of the UMAP figure for more accurate description. We believe the overlap in composition space is expected and it may suggest that LLMs favor known compositional patterns as a result of stability-focused generation.
> >
> > 3.3 Novelty Category Definition Clarification: Yes. Novel Polymorphs means structures with formulas that exist in Materials Project but no structural match, which represent new crystal arrangements of known compositions. We have detailed the definition of each category in Appendix F.2.
> >
> > 4. We added a clarification in Section 5.3 noting that DiffCSP demonstrated superior performance compared to USPEX.

---

> > > ### Comment · Reviewer_7o1E · 2026-01-15
> > >
> > > I appreciate the response to my concerns! Things are looking better, but I still am a bit confused by some of your definitions...
> > >
> > > - In section 4.3.2 you mention that your method "achieving 37.59% metastability rate (Ed < 0.0 eV/atom)". DFT calculations with decomposition energies below 0.0 (after the judicious and careful application of the proper MP2020 corrections) are something that I would call Stable. You also refer to this as stable in the S.U.N. definition & in the table. Please clarify.
> > >
> > > The authors must properly and explicitly define the following and clearly indicate this in the text. The best way would be with some summary table in the appendix:
> > > - The definition of stability in Figure 5 and with respect to which hull (I assume `2023-02-07-ppd-mp`, but you are comparing uniqueness and novelty to something else than the hull, which is unusual...)
> > > - The definition of unique and novel for every dataset and model including your own, i.e. both settings and database to compare to.
> > > - Changing the name Stability in Table 1 to something like "Stability % Relative to Metastable," or "SabRelMeta," etc. Having both of these definitions in the same text is ripe for problems in comprehension. (I understand they are comparing to `CrystalTextLLM-70B` which does this analysis so it is necessary.) Another suggestion would be to use different colors for each of the definitions of stability, metastability, etc.
> > > - The definitions of stability, metastability, etc. for every other model you are comparing with.
> > > - The corrections that were applied by every method in the corresponding computations.
> > >
> > > If you make this clear and reference it plainly in the main text with open-soruced structures that back it up, I believe you will have done your due diligence in reporting such a high S.U.N. rate. I am taking this so seriously because your model, according to your reported metrics, may be the most effective model for producing stable and novel structures. However, your analysis has contained confusing definitions and errors in the text leave me concerned about (a) whether the analysis was done correctly and (b) whether I accurately understand what you are reporting. Please take extreme care in reporting these results and making every use of the words "stable," "metastable," "unique," or "novel" very clear!

---

> > > > ### Author Response · Authors · 2026-01-22
> > > >
> > > > We are sincerely grateful for all the valuable feedback, detailed questions, and constructive suggestions. We fully agree that precise terminology and explicit metric definitions are essential. In the revision, we reorganized the evaluation sections and expanded the appendix to make all use of "stable", "metastable", "unique", and "novel" unambiguous, while also keeping comparisons to prior work as fair as possible.
> > > >
> > > > **Clarification: what we mean by S.U.N. and how we report it**
> > > >
> > > > We emphasize that **DFT-verified stability is the primary metric and the primary evidence** for our claims. The S.U.N. rate is included only as a supplementary quality indicator because it (i) collapses multiple quality dimensions into one number and (ii) is highly sensitive to the choice of reference set used for novelty, which differs across methods. We have explicitly discussed in Appendix D.4 why direct cross-method S.U.N. comparisons are inherently limited when novelty is computed against different reference sets.
> > > >
> > > > In the revised manuscript, we report S.U.N. in two contexts:
> > > > - Main text Section 4.3.1, Table 2 (Original Section 4.3.2, Table 2): S.U.N. computed using DFT-verified stability (with CHGNet pre-relaxation followed by DFT relaxation), matching the evaluation spirit of baseline comparisons as closely as possible. We report both metastability and stability in Table 2.
> > > > - Appendix (large-scale screening, MLIP-based, Appendix G.4.2, Table S9): S.U.N. computed using CHGNet-predicted metastability ($E_d < 0.0$ eV/atom) due to computational resource constraints. This is explicitly labeled as `MLIP-based’ and not used as the primary claim.
> > > >
> > > > We take seriously the responsibility of accurately reporting the results. For reproduction purposes, we have provided the source code and generated structures in Appendix A, and prompts used in the experiments in Appendix B.
> > > >
> > > >
> > > > **Clarification: "metastability" vs "stability":**
> > > >
> > > > To ensure terminological precision and consistency throughout the manuscript, we have reorganized Section 4.2 and Appendix D to comprehensively clarify all evaluation metrics. We have also revised the manuscript to ensure all terminology is used consistently:
> > > > - "Stable" exclusively refers to DFT-verified results ($E_d < 0.0$ eV/atom, VASP).
> > > > - "Metastable" exclusively refers to MLIP predictions (CHGNet by default, unless otherwise specified).
> > > > - All thresholds are explicitly stated each time metastability is mentioned.
> > > >
> > > >
> > > > **Requested “explicit summary table” of definitions**
> > > > We fully agree and have implemented this recommendation. In the revision we added a summary table of definitions (Table S1 in Appendix D.6) to summarize all definitions, datasets, and corrections.
> > > >
> > > > **Discussion of why our framework improves stability:**
> > > >
> > > > We added a discussion highlighting that stability depends on both composition and geometry. Diffusion-style baselines may generate chemically unfavorable compositions or prototypes with plausible geometry but poor thermodynamics. Our evolutionary framework uses an MLIP oracle to guide selection and variation, thereby filtering out unfavorable candidates during search and concentrating computation on more promising regions of chemical and structural space. We incorporate this perspective into the revised conclusion (Section 6).

---

### Comment · Action_Editor_WCAK · 2025-10-31
**Update of the Review Process**

Dear Authors,

Thank you very much for submitting your work to TMLR, and sorry for the long waiting time after submission. Here is an update on the current status.

After being assigned to this submission, I found that there were no qualified reviewers in the current TMLR reviewer pool for your manuscript. Therefore, I have actively invited many external reviewers with relevant expertise. Recently, we have finally secured three expert reviewers with backgrounds in learning-based crystal structure or material generation for your submission. In addition, an emerging reviewer with a background in LLM has also been assigned to this work.

As a result, the review deadline has been automatically extended by the system to November 14, which is two weeks after the last reviewer's confirmation. I expect to receive three high-quality reviews from the expert reviewers by the deadline. Once three reviews are obtained, the emerging reviewer will be unassigned from this work in accordance with TMLR policy.

Thank you for your understanding of this arrangement. Please feel free to reach out if you have any questions or concerns regarding the review process.

Best regards,

AE

---

### Author Response · Authors · 2026-01-22
**Summary of Change**

To address all reviewer concerns, we have revised the manuscript and reorganized the structure for clarity. All revised content is marked blue in the manuscript. Below we provide a summary of major additions and revisions:

- **Evaluation Metrics (Section 4.2, Appendix D; 7o1E)**: Added Section 4.2 consolidating evaluation metric definitions with explicit thresholds and reference databases. Expanded Appendix D with details on stability, metastability, novelty, and S.U.N. methodology, including Table S1 summarizing all definitions, databases, and energy corrections.

- **S.U.N. Methodology Clarification (Section 4.3.1, Appendix D.4; 7o1E)**: Clarified S.U.N. rate computation methodology, added visual comparison schematic differentiating our calculation from baseline methods, and positioned S.U.N. as a supplementary metric while maintaining DFT verification as the gold standard.

- **Broader Impact (Section 6; 7o1E)**: Extended discussion on broader impact and limitations.

- **Bulk Modulus Controllability (Appendix E; 4z2W)**: Added targeted bulk modulus optimization experiments demonstrating property-guided generation and multi-objective optimization capabilities.

- **Cross-Family LLM Comparison (Appendix G.4.2; 4z2W)**: Added evaluation on 6 LLMs across multiple model families (GPT, Grok, DeepSeek, Claude) demonstrating cross-model generalization.

- **Larger-Scale Generation (Appendix G.5; 7o1E)**: Extended experiments to population size K=500 with DeepSeek-Reasoner, with novelty and stability analysis provided.

- **Carbon Footprint (Appendix G.6; 7o1E)**: Added environmental impact analysis quantifying computational resources and carbon emissions.

- **Novelty Verification Against Materials Project (Appendix H.2; 7o1E, dFDA)**: Extended novelty verification against Materials Project database with detailed breakdown of compositional novelty, structural novelty, and novel polymorphs.

- **Failure Mode Analysis (Appendix I; dFDA)**: Added comprehensive failure mode analysis categorizing different types of generation failures, quantitatively analyzing static chemical inconsistencies and structural validity issues.

- **Mutation and Crossover Analysis (Appendix J; 4z2W)**: Added detailed analysis of mutation and crossover operations in evolutionary search.

---

### Decision · Action_Editor_WCAK · 2026-02-08

**Recommendation:** Reject

**Additional Comments:**

**Initial Review**: Since there were no suitable reviewers in the TMLR reviewer pool for this submission, three external experts with backgrounds in learning-based crystal structure generation or material generation have been invited to review this work. After the initial review and rebuttal, the reviewers had mixed opinions and did not reach a consensus regarding this work. Reviewer 4z2W leaned toward accepting this work, though with remaining concerns and suggestions for further improvement. Reviewer dFDA recommended rejecting this work and believes the revised manuscript raises more questions than it answers regarding the effectiveness of the proposed approach. Reviewer 7o1E wanted to engage in an extra round of discussion with the authors and indicated they could consider supporting acceptance if the authors could adequately address the remaining concerns.

**Extra Round of Discussion:** Since I do not have sufficient expertise in material generation to make a fair decision alone, I have asked the reviewers for help. Two reviewers (Reviewer 7o1E and Reviewer 4z2W) have engaged in an extra round of discussion with the authors to further adjust their official recommendation. It should be noticed that this is a completely voluntary task beyond the reviewers' original thorough review, and I truly appreciate all reviewers' time and effort spent reviewing this submission for TMLR.

**Final Recommendation:** The authors have actively participated in all the rebuttals and discussions, and have continually revised and improved the paper. However, after two rounds of thorough discussion, two reviewers (Reviewer dFDA, Reviewer 7o1E) still believe the claims made in this paper are not supported by convincing and clear evidence, and lean toward rejection. Therefore, I have to recommend **rejecting** this work at this time. Based on the reviewers' comments, I also recommend an **optional resubmission of major revision** for this work.

The reviewers have provided detailed and valuable suggestions in their final official recommendation. Since these suggestions are not visible to the authors, I attached them below and hope they would be useful for the next revision of this paper.

**Reviewer dFDA:**

I thank the authors for conducting additional experiments in response to the reviews. However, despite these efforts, the revised manuscript raises more questions than it answers regarding the effectiveness of the proposed approach. For instance:

1. To match the 10k generations used in prior work for a fair comparison, the authors employ the DeepSeek Reasoner (approximately 600B parameters) to generate about 8.6k samples. This model is substantially larger than the baseline (CrystalTextLLM-70B), yet no explanation is provided as to why the same number of generations could not be produced using Llama-3.1-70B.

2. In Table S3, instead of directly applying the proposed evolutionary algorithm to CrystalTextLLM-70B (a carefully fine-tuned Llama-70B model), the authors choose to fine-tune Llama-3.1-70B from scratch. No justification is given for not using CrystalTextLLM-70B directly. This raises concerns about the credibility and fairness of the reported experiments (e.g., whether sufficient hyperparameter tuning was performed for the newly fine-tuned model).

3. The authors additionally report results from newer models, but do not specify how many generations correspond to each model, making it difficult to interpret or compare the results.

Overall, while the proposed approach appears sound and could be of interest to the machine learning and materials science community, the authors do not sufficiently justify its effectiveness. As presented, the experimental results do not adequately support the claims, leaving the reader with confusion and lingering concerns about the robustness of the conclusions.


**Reviewer 7o1E:**

Most of Reviewer 7o1E's very detailed and valuable suggestions can be found in their public comments below. In a priviate comment to AE regarding the final recommendation, Reviewer 7o1E provided additional suggestions:

However, I think that as a result of all this back and forth the paper is extremely confusing to read now. I acknowledge that my responses have contributed to this result, but the paper should make sense to someone who didn't go back and forth with the authors on a review like this. They have multiple definitions of stability in the main text rather than converting results from other papers into to a consistent one. They have significant sections of text explaining caveats to their calculations in comparison with baselines.

It's a tough call because the authors have engaged in good faith efforts to improve the paper, albeit often fraught with confusing additions. At this point I would lean reject but with a strong note that the results are good and the paper should be published, but just after more effort is taken to make tables and results read consistently throughout the main text. The text should have limited caveats, not multiple per table/figure like in the current state.


**Reviewer 4z2W:**

*Comments after the Initial Review:*

I thank the authors for their detailed responses to my questions, which include some additional experiments. I also carefully reviewed the responses to other reviewers. I lean towards accepting this work, primarily because the authors have answered and clarified most of the concerns posed by the reviewers, including the fairness of comparison of metrics, data leakage, scalability, analysis of LLM-generated mutations/crossovers, and failure analysis. Additionally, the authors have addressed and corrected many of the requested changes. Overall, the methodology is novel, and orthogonal to standard training and fine-tuning pipelines for crystal generation. The extent of the experiments demonstrates the reliability of such evolutionary strategies.

However, further concerns and suggestions that the authors could consider focusing on are provided below.

1. As highlighted by Reviewer 7o1E, **direct performance comparison of the proposed method with traditional evolutionary approaches like USPEX, XtalOpt, GASP, etc. is missing**.: It is unclear if the usage of pre-trained LLMs offers a clear advantage without this additional analysis. For instance, the authors claim that traditional crossover techniques and mutation techniques (e.g. element substitution, atomic displacement, etc.) limit exploration, and that LLM-guided reproduction overcomes this issue. This statement would be more relevant if compared with traditional algorithms.

2. Unlike generative approaches like CDVAE, DiffCSP, FlowMM that can generate crystals starting from noise, MatLLMSearch's capacity seems to largely depend on the initial set of seed structures. It is **not clear if the authors averaged the performance over multiple seeds** (i.e., multiple sets of initial structures), which can also give an estimate of the variance. The provided code specifies a random seed while loading the structures - so it would be great if the authors clarify this.

3. The authors clarified the bulk modulus calculation protocol. However, for evaluation, it should be easy to **compute the bulk modulus using DFT** and verify the results, which the authors have not done.

4. I would also encourage the authors to consider aligning their evaluation of generated structures with LeMat-GenBench, which provides a more recent and standardized set of evaluation protocols and metrics.

*Comments after the Extra Round of Discussion:*

I stay with my current recommendation of leaning towards acceptance, mostly because of the novelty in framing the problem and other strengths mentioned in my original review (DFT validation, ablations, etc.). I acknowledge the concerns pointed out by other reviewers and believe they can be addressed in the revised version. I have provided some thoughts below. I hope this helps.

1. My initial review mainly focused on the underlying methodology, the motivation behind LLMs and the proposed evolutionary approach, crossover details, and property optimization. The authors have answered most of my concerns and have promised some changes in the revised version.

2. There were some concerns regarding how the metrics were calculated and reported. For a fair comparison, it is expected that 10k structures are sampled from the proposed and baseline models, followed by computation of validity, stability and S.U.N metrics. I assume that there are computational bottlenecks in running evolutionary search for more than 10 iterations (this resulted in a much lower number of generated crystals).

3. Besides, unlike previous diffusion/LLM-based models, where sampling is performed independently each time, there is an iterative component where the children of one iteration can influence the output of the next iteration. Direct comparison of metrics with existing models appears a bit tricky. While the authors have clarified the discrepancies pointed out, I'm not entirely sure how their stability and SUN score are significantly better than the baselines.

4. Additional comments: As mentioned in one of my comments, the authors have not averaged the performance over different seeds, but they replied that their preliminary experiments with different seeds indicated no change in conclusions. Not performing DFT verification for bulk modulus is understandable. Further, I am generally satisfied with the crossover analysis, although direct comparison with classical evolutionary approaches might give a lot more insights.

**Audience:**

Yes

**Audience Explanation:**

All reviewers believe some individuals in TMLR's audience could be interested in the findings of this paper.

**Claims And Evidence:**

No

**Claims Explanation:**

This work proposes MatLLMSearch, an evolution-guided large language model (LLM) based framework for crystal structure discovery. The key idea is to use LLMs within an evolutionary workflow to conduct crystal structure generation (via implicit crossover and mutations), prediction, and objective-based optimization. The results show that MatLLMSearch can outperform specialized models such as CrystalTextLLM on some crystal structure generation tasks. In addition, MatLLMSearch can also adapt to other material design tasks, such as crystal structure prediction and multi-objective optimization of properties.

One reviewer believes the claims made in this paper are supported by convincing and clear evidence, but the other two reviewers do not think so. Please refer to the comment below for more details.

**Resubmission Of Major Revision:**

The authors may consider submitting a major revision at a later time.